

# Evaluating and developing a model of specific degradation using geospatial analysis for sediment erosion management in South Korea

Woochul Kang[1]
Pierre Y. Julien[2]

[1] Department of Land, Water and Environment Research. Korea Institute of Civil Engineering and Building Technology (KICT)., Goyang-si 10223, Gyeonggi-Do, Korea
[2] Dept. of Civil and Env. Engineering, Colorado State Univ., Ft. Collins, CO 80523

*Correspondence to:* Woochul Kang (kang@kict.re.kr)

**Abstract**

The South Korean Peninsula is subject to hydrological extremes, and 70% of its terrain is mountainous, with sharp ridges and steep valley flanks. Recently, rapid urbanization has created an emerging demand for large-scale water resources, such as dams and reservoirs. Accordingly, complicated sediment-related problems have become an issue, with abundant soil loss during typhoons transported to the reservoirs, and downstream, riverbed degradation is caused by intercepting sediment. Thus, a reliable approach is required for predicting sediment yields of soil erosion and sedimentation. In this study, the specific
degradation (SD) of 62 stream-river watersheds and 14 reservoir watersheds were calculated from field measurements of sediment concentration and deposition. Estimated SD ranged between 10 and 1,500 tons·km$^{-2}$·yr$^{-1}$. Furthermore, existing empirical models of sediment yield are insufficient for predicting specific degradation upstream of the reservoirs; therefore, a new model was developed based on multiple regression analysis and model tree data mining of 47 watersheds (~75% national land cover). Accuracy of the developed model was enhanced with the following significant parameters: (1) drainage area, (2)
mean annual precipitation, (3) percent urbanized area, (4) percent water, (5) percent wetland and water, (6) percent sand at effective soil depths of 0–10 cm, (7) slope of the hypsometric curve, and (8) watershed minimum elevation. Additionally, erosion maps from the revised universal soil loss equation (RUSLE) were generated to validate model variables and further understand the sediment regime in South Korea. The gross erosion results for 16 ungauged watersheds were used to validate the empirical model by comparing sediment delivery ratios of other references. The modeled meaningful parameters were
examined via remote sensing analyses of satellite and aerial imagery and revealed the features affecting erosion and sedimentation with an erosion loss map at 5-m resolution. Vulnerable areas of soil loss, including construction sites, and croplands, as well as sedimentation features, such as wetlands and agricultural reservoirs, were highlighted.

**Keywords** Specific degradation, sediment yield, geospatial analysis, remote sensing, resolution effect, revised universal soil
loss equation (RUSLE), data mining, multiple regression

## 1 Introduction

The process of soil erosion can be classified into three chronological stages: detachment, transport, and deposition. Some initial terminology related to erosion and sediment must first be clarified. "Gross erosion" refers to all erosion within a watershed. Commonly, erosion from water is composed of sheet (i.e., inter-rill), rill, gully, and instream erosions. When soil particles
detach, they become part of the flow. At a specific location, the gross erosion material transported downstream is known as "sediment yield." When the transport capacity of runoff is insufficient, deposition can occur within or even before reaching the stream. The "sediment delivery ratio" (SDR) can be calculated as the ratio of sediment yield to gross erosion (Julien, 2010). Although the revised universal soil loss equation (RUSLE) considers only sheet and rill erosions, it has been widely used to



estimate the overall sediment budget of catchments (Zhao et al., 2012; Kamaludin et al.,2013). The sediment yield is commonly

measured at a river gauging station, or as accumulation in a reservoir. The term "specific degradation" (SD) refers to the ratio

of mean annual sediment yield to the watershed area (Julien, 2010).

In South Korea, most mountains and valleys (~65% of the total land area) are in a stage of old age, with deep valleys in the

upstream regions and broad alluvial flat plains downstream (Fig. 1 (a)). Most of the mountainous watersheds are forested, and

the plains are commonly cultivated as paddy fields, effectively functioning as small reservoirs for holding water during the

summer (Yoon and Woo, 2000). Furthermore, ~17,000 reservoirs have been constructed to meet regional water demands (Fig.

1 (b)). These conditions tend to reduce large-scale problems related to erosion and sedimentation. However, numerous

concentrated and local issues are ongoing (Kang et al., 2021). For example, abundant soil loss can occur during typhoons, and

the soil particles generated upstream induce sediment problems in reservoirs. Further downstream of the reservoir, riverbed

degradation can be caused by intercepting sediment. Additional local issues can include landslides, aggradation, and floodplain

sediment deposition.

Accordingly, many scholars have suggested that an accurate method for predicting sediment yield under local conditions is

required for sustainable management (Scherer, U., & Zehe, E. 2015; Gao et al., 2016; Kang et al., 2019). To this end, several

methods have been applied in South Korea, including empirical models developed using regression analysis (Ryu and Min,

1975; Ryu and Kim, 1976; MOC, 1992), and data mining techniques to derive new empirical equations of sediment discharge

(Zhu et al., 2007; Kang and Jang, 2020; Kang et al., 2021). The existing formulas have produced highly variable results,

because the underlying models were designed with different conditions and purposes in mind, and in some cases, were based

on a relatively low number of observations. In particular, appropriate variable selection is fundamental for modeling sediment

yield, which requires proper understanding of the influential mechanisms (Vente et al., 2011). Furthermore, upland gross

erosion of a catchment and downstream sediment yield should be considered together to calculate the SDR and better

understand the processes of sediment transport (Julien, 2010). The magnitude of the SDR is influenced by many factors, such

as geomorphological, environmental, and artificial ones (Walling, 1983); however, spatiotemporal limitations present

challenges, including uncertainties arising from data-scarcity and general issues of regional applicability (Gao et al., 2017).

Thus, gross erosion and SDRs are not often used in practice by geomorphologists and river managers, save for some specialized

purposes (Benavidez et al., 2018; Swarnkar et al., 2018).

In this study, an abundance of field data was used to estimate specific degradation using a modified Einstein procedure and

flow duration-sediment curve method. Subsequently, empirical models were developed from the estimated SD of 47

watersheds, and 51 parameters derived from GIS analyses and field surveys. The results of this study suggest new uses of the

SDR for validation of the proposed empirical model, and the SDRs from estimated gross erosion of ungauged watersheds by

RUSLE were compared with modeled SD estimates. Additional geospatial analyses using erosion maps, satellite, and aerial

imagery were used to identify meaningful erosion and sediment features and explain their effects on the soil erosion process.

Because the grid size of parameters for RUSLE should not be arbitrary, the distortion of gross erosion was examined at three

different spatial resolutions.

## 2 Materials and Methods

### 2.1 Specific degradation of rivers, streams, and reservoirs

There were 62 sediment data points compiled for rivers, streams, and 14 reservoirs across five main rivers ((Fig. 1 (a)). Most

of the major reservoirs with multipurpose dams are located in the mountainous regions (Fig. 1 (b)). When water enters a





reservoir, the flow velocity decreases, flow depth increases, and sedimentation occurs as a result of the overall decreased transport capacity of the stream. The amount of deposited sediment in the reservoir is dependent on sediment production from upland erosion, the rate of transportation, and the mode of deposition (Julien, 2010). Korea Water resource corporation (K-water), the public body in charge of the construction, operation, and management of facilities for water resources, has published a sediment survey report of the multipurpose and storage dams of several reservoirs every ten years. The sediment deposition

rate ($V_r$, m$^3 \cdot$km$^{-2} \cdot$yr$^{-1}$) can be derived from field measurements of the reservoir capacity. In the present study, $V_r$ of the reservoir was calculated according to Eqs. (1) and (2):

$$V_r = (V_m - V_i)/(A \times t) \quad , \tag{1}$$

$$SD = (V_r \times \rho_{md})/T_E \tag{2}$$

where $V_m$ is the measured reservoir capacity from the impoundment of water (m$^3$), $V_i$ is the initial capacity of the reservoir at

impoundment (m$^2$), $A$ is the catchment area (km$^2$), $t$ is the time from the impoundment of water to measurement (yr), $SD$ is the specific degradation (tons$\cdot$km$^{-2} \cdot$yr$^{-1}$), $T_E$ is the trap efficiency (%), and $\rho_{md}$ is the dry specific mass of the sediment deposition (tons$\cdot$m$^{-3}$). The reservoir trap efficiencies for multiple-purpose dams in South Korea are commonly > 96%, and a dry specific mass of 1.1 tons$\cdot$m$^{-3}$ was assumed when field measurements were unavailable (MOC, 1992). In this study, the most recent survey data were used, and the estimated reservoir SD ranged from 200 to 1,800 tons$\cdot$km$^{-2} \cdot$yr$^{-1}$ (Table 1). For the Imha,

Miryang, and Buan reservoirs, the survey report concluded that the estimated total sediment data for the reservoir was unreliable; and therefore, provided identical results as the design value for the estimated sediment deposit rate.

In terms of sediment data, stream and river watersheds were classified according to their location either up- or downstream of major reservoirs, respectively (Fig. 1 (a)). A total of 2,432 suspended sediment measurements were collected across 62 gauging

stations. First, the total sediment load was estimated using a Modified Einstein Procedure (MEP; detailed methods were reported by Julien (2010)). The relationship varied between the estimated sediment load, and the discharge at the time of sediment measurement in streams and rivers (Fig. 2 (a)). Notably, total sediment discharge was higher at low discharge rates in streams. The mean channel characteristics from sediment measurement data are shown in Fig. 2 (b) and (c), and the results reflect the different characteristics of river and stream watersheds. Stream watersheds maintain a V-shaped valley in upland

areas, with lower discharge and smaller cross-sectional areas than rivers. Gauging stations of rivers confirmed that they maintain relatively deep and wide channels at low elevations with mild slopes, ultimately indicating that they are in the transport zone, with long sand-beds. With the estimated total sediment load, the annual SD values were calculated using the flow duration–sediment rating curve (FD-SRC) method (Table 2). Consequently, 15 results with low sample numbers (sediment measurements < 15) and one unreasonably high result of the 62 river, stream, and reservoir stations were removed

from modeling analyses.

## 2.2 Modeling specific degradation

Models of sediment yield were developed using the estimated SD data and 51 GIS-based watershed parameters by multiple regression analysis and data mining. The parameters considered were classified as geomorphological, climatological, anthropogenic, or pedological factors (Table 3). Geomorphological parameters were acquired during watershed delineation

according to a 5 m × 5 m resolution digital elevation model (DEM) provided by the National Geographic Information Institute. Climatological variables were based on 60 points of precipitation and rainfall erosivity data from the Korea Meteorological Administration. The continuous raster data of mean annual precipitation and rainfall erosivity at 5-m resolution was generated by the ordinary kriging with the exponential semivariogram model which is reliable for rainfall data. Land cover type related to anthropogenic parameters was derived from 5 m × 5 m resolution raster data, classified by a hybrid method of the Korean

Ministry of Environment from the Landsat TM, IRS 1C, SPOT5, and Arirang satellite images (Me, 2002). Pedological





variables were obtained from the National Institute of Agricultural Sciences and SWAT-K developed by the Korea Institute of Construction Technology. The detailed processes and information of all parameters were reported by Kang (2019).

The regression model based on the RUSLE structure for SD was represented according to Eq. (3):

$$\ln SD_i = \beta_0 + \beta_1 lnX_{i1} + \cdots + \beta_{p-1} lnX_{i,p-1} + \varepsilon_i \tag{3}$$

where $\beta$ is the regression coefficient, $X$ is the explanatory variable, $\varepsilon_i$ is the error term, and $p$ is the number of explanatory variables. For the developed model, the confidence intervals were calculated according to Eq. (4):

$$SD_h \pm t\left(1 - \frac{\alpha}{2}; n - p\right) s\{SD_h\} \tag{4}$$

where $\alpha$ is the level of significance (assumed as 0.05 for 95%) and $s\{SD_h\}$ is the standard deviation. Data mining can be used to effectively derive explicit formulas by separating the data into subtrees. Branches are divided when the standard deviation

reduction is maximum (Quinlan, 1992) according to Eq. (4):

$$\begin{matrix} standard \ deviation \\ reduction \end{matrix} = \begin{matrix} standard \\ deviation \end{matrix} (T_i) - \sum_i \frac{|T_i|}{|T|} \times \begin{matrix} standard \\ deviation \end{matrix} (T_i) \tag{5}$$

where $T$ is the set of total samples of the dependent variable and $T_i$ is the set of subsamples of the dependent variable divided by the sub-intervals. The minimum data under the model tree classification conditions were assumed to be four for multiple regression analysis, and the standard deviation reduction was set to 5% to avoid unnecessary and excessive results.


RUSLE was used to validate the developed models by remote sensing analyses of erosion and sedimentation using satellite and aerial imagery at different resolutions. Wischmeier and Smith (1965) developed the universal soil loss equation (USLE) using data from 10,000 test plots of agricultural areas in the U.S., later modifying this equation in 1978 (Eq. 6):

$$A = RKLSCP \tag{6}$$

where $A$ is the average annual soil loss, $R$ is the rainfall erosivity factor, $K$ is the soil erodibility factor, $L$ is the slope length factor, $S$ is the slope steepness factor, $C$ is the cropping management factor, and $P$ is the conservation practice factor.

To validate the developed model, the calculated SDRs of ungauged watersheds from gross erosion and predicted SD were compared with ranges acquired from the literature. In this study, $R$ values at 60 stations were calculated according to Eq. (7):

$R = \sum E \cdot I_{30}, E = \sum e \cdot \Delta P, e = 0.29[1 - 0.72 \exp(-0.05 \cdot I)] \tag{7}$

where $R$ is reported in ($10^7 \text{J} \cdot \text{ha}^{-1} \cdot \text{mm}^{-1} \cdot \text{h}^{-1}$), $I_{30}$ is the maximum 30-min rainfall intensity (mm·h⁻¹), $E$ is the total storm kinetic energy ($10^7 \text{J} \cdot \text{ha}^{-1}$), $\Delta P$ is the increase in rainfall during the rainfall interval (mm), $e$ is the estimated unit rainfall kinetic energy (MJ·ha⁻¹·mm⁻¹), and $I$ is the rainfall intensity (mm·h⁻¹). The soil erodibility factor ($K$) was calculated for each soil group according to Eqs. 8–10 (Wischmeier et al., 1971):

$K = \frac{0.00021 \cdot M^{1.14} \cdot (12 - OM) + 3.25(C_{soilstr} - 2) + 2.5(C_{perm} - 3)}{100} \tag{8}$

$M = (m_{silt} + m_{vfs}) \cdot (100 - m_c) \tag{9}$

$OM = 1.72 \cdot orgC \tag{10}$

where $K$ is the soil erodibility factor ([ton·acre·h]·[hundreds of acre·foot-tonf·inch]⁻¹), $M$ is the particle size parameter, $OM$ is the percentage of organic matter, $C_{soilstr}$ is the soil structure code used for classification (1–4), $C_{perm}$ is the profile permeability

class (1–6), $m_{silt}$ is the percentage of silt, $m_{vfs}$ is the percentage of fine sand, $m_c$ is the percentage of clay, and $orgC$ is the percentage of organic carbon content of the layer.

For the topographic factors ($LS$), field measurements yielded the best results; however, various functional and practical GIS algorithms have also been developed for this purpose. Modeled results of gross erosion from GIS analyses have shown to be dominated by spatial heterogeneity, represented by $LS$ in the RUSLE (Risse et al., 1993; Thompson et al., 2001; Wu et al.,

2005), implying that the resolution of the DEM used is directly related to the accuracy of topographic factors, and thus, soil



loss (Thompson et al., 2001). Moore and Wilson (1992) suggested a simple equation using a unit contributing area to estimate the LS factor (Eq. 11):

$$LS = \left(\frac{\lambda}{22.13 \; for \; SI \; unit}\right)^m \cdot \left(\frac{\sin(\theta)}{0.0896}\right)^n \tag{11}$$

where $\lambda$ is the flow path length (m), $\theta$ is the slope angle (in degrees), and $m$ and $n$ are constants. Moore and Burch (1986)
suggested that $m$ is 0.4 (range, 0.4–1.6) and $n$ is 1.3 (range, 1.2–1.3). The flow path length ($\lambda$) is the distance from the point of origin of overland flow to the point where the slope decreases sufficiently to allow deposition to begin, or where the runoff enters a well-defined or constructed channel (Wischmeier & Smith, 1978). *Flow Accumulation* is one of the most common GIS tools for estimating flow path length, and when these values are multiplied by the DEM resolution, the distance travelled by a drop of water before reaching that particular cell is derived (Parveen and Kumar, 2012; Barriuso Mediavilla et al., 2017);
however, this method is limited because the higher gross erosion results occur at the convergence of a catchment. Another option for estimating the *LS* was reported by Hickey (2000) and Van Remortel et al. (2001), based on single flow direction algorithms. These methods considered the downhill slope angle with a directional component, and non-cumulative slope length for high points (Van Remortel et al., 2004), thus diminishing the limitations from the unit contributing area (Galdino et al., 2016). In this study, (1) Moore and Wilson's method based on unit contributing area and (2) Van Remortel's method based on
the flow path and cumulative cell length portion (FCL) were used to estimate *LS*. The cropping management factor (*C*) and conservation practice factor (*P*) were referenced from the Ministry of Environment's regulations and one additional source (ME, 2012; Kim, 2016), and their values are listed in Tables 4 and 5, respectively. *P* is dependent on the conditions of land use and slope, where the percent slope was derived from the DEM, and land use was based on the land cover raster. Although the advantages of calculating RUSLE with GIS are numerous, variable grid/cell sizes result in different soil erosion results.
Molnár (1997) concluded that a large grid cell size (low spatial resolution) underestimated soil loss, and Julien and Frenette (1987) used a correction factor to extend the applicability of the RUSLE to larger watersheds. In this study, the resolution effects on the results of gross erosion were examined. Wu et al. (2005) suggested that sampling methods do not affect USLE application; therefore, the nearest-neighbor technique was applied for downscaling the DEM. Both land use and soil classification were downscaled by a majority function to determine the new cell value based on the most common values
within the corresponding cells. Finally, the developed regression model was validated for ungauged watersheds in South Korea using the SDR, as defined by Eq. (12):

$$SDR = A_T/SY \tag{12}$$

where $SY$ is the sediment yield (estimated SD from the developed model) and $A_T$ is the gross erosion from the watershed (results of RUSLE). Mapping erosion at different spatial resolutions using the RUSLE can also help identify meaningful
erosional and sediment features.

## 3 Results

### 3.1 Modeled specific degradation

The SD results for rivers and streams decreased with watershed area, with river values being the lowest of the three water
bodies analyzed. As expected, SD decreased from mountain streams to river valleys, suggesting that upland erosion occurs mainly in the upstream portions of the watersheds, while sedimentation is observed in the reservoirs and alluvial rivers downstream. The SD across reservoirs was relatively constant, with a mean value of 850 ton·km$^{-2}$·yr$^{-1}$. The models for predicting sediment yield using multiple regression and a model tree analyses were developed with 47 results of SD for rivers and streams and 51 watershed parameters derived from GIS analyses.

Existing Model: $SD = 2.45 \times 10^{-7} A^{-0.04} P^{1.94} U^{0.61} W^{-0.64} Sa^{1.51} Hyp^{1.84}$ $\tag{13}$





Proposed Model: $SD = 1.75 \times 10^{-5} A^{-0.07} P^{2.23} U^{0.4} WW^{-1.04} Hyps^{-0.42}$ (14)

Previously, Kang et al. (2019) developed empirical models for watersheds in South Korea using only 28 SD values for rivers. These existing models were validated by comparison with other empirical models of sediment yield in South Korea, and the most accurate model was derived (Eq. 13); however, it did not show sufficient performance for the SD of streams (RMSE =

130.1 tons·km$^{-2}$·yr$^{-1}$, NSE = 0.284; Fig. 4 (a)). In the proposed model using multiple regression analysis and 47 results of SD for rivers and streams (Eq. 14), the most important parameters were watershed area (*A*), mean annual precipitation (*P*), percent urbanized area (*U*), percent wetlands and water area (*WW*), and slope of the hypsometric curve (*Hyps*). The proposed model based on multiple regression analysis predicted the SD of streams and rivers more accurately (Fig. 4 (b)). Validation was conducted using the additional reference from MOC (1992), and all the points were within the approximated 95% confidence

intervals (RMSE = 88.8 tons·km$^{-2}$·yr$^{-1}$, NSE = 0.516). Additionally, the issue of multicollinearity for the proposed model was ruled out by checking the variance inflation factor (VIF), according to Eq. (15).

$$\mathrm{VIF}_i = \frac{1}{1-R_i^2}$$ (15)

Multicollinearity was ruled out, as none of the VIF values were > 10 (Table 6; Kutner et al., 2004).

Additionally, the derived empirical model using data mining was proposed with five meaningful parameters (Eqs 16 and 17):

$LE \leq 71.88$ m; M1 $= 9.3 \times 10^{-9} \times P^{2.8} \times SA010^{0.48} \times U^{0.71} \times W^{-0.43} \times LE^{0.036}$ (16)

$LE > 71.88$ m; M2 $= 2.5 \times 10^{-3} \times P^{1.4} \times U^{0.29} \times W^{-0.27} \times LE^{0.056}$ (17)

Fig. 5 (a) delineates the divided groups according to the model's suggestion and Fig. 5 (b) shows the classified SD for the rivers and streams by their locations.

Both the RMSE (58.8 tons·km$^{-2}$·yr$^{-1}$) and NSE (0.56) indicated a superior performance compared to the multiple regression analysis model. According to the results, the data mining model tree well distinguished the SD values of streams and rivers according to the lowest elevation (*LE*). The proposed data mining model had a similar to the other, where the lowest elevation and slope of the hypsometric curve were used in both proposed models as relief aspects. Interestingly, the regression model included the percent of sand at effective soil depths of 0–10 cm, possibly indicative of the difference between sand bed rivers

with fine suspended material, and gravel and cobble-bed streams.

### 3.2 Model and geospatial analyses with RUSLE

The estimated average gross erosion values obtained using the two approaches to *LS* factor calculation, at three different spatial resolutions and across the 15 watersheds, are shown in Table 7.

Many researchers have suggested that the method for estimating the *LS* factor, and DEM resolution, are directly related to the

accuracy of topographic factors. In this study, the UCA and FCL methods were used to estimate slope length and steepness. The main difference between the two methods was that flow accumulation does not consider the cutoff point where sediment will be deposited in UCA. Conversely, the FCL method accounts for deposition issues by evaluating changes in slope; however, it does not consider convergence flow or channel networks (Zhang et al., 2013). South Korea's steep montane topography results in most watersheds maintaining smaller areas compared to other continental regions. Moreover, 65% of the precipitation

falls during the summer rainy season, creating a correlated peak in sediment transport. Accordingly, there are many flow convergence points within these small watersheds; as a result, the FCL method yields a lower LS value for concave areas. When assessing spatial resolution, the results were corroborated by those of other studies (Molnár, 1997; Wu et al., 2005), in that gross erosion slightly decreased with higher DEM resolution because the cut-off points in the 5 m-resolution DEM were more apparent.





Since there was a divergent trend observed between SD values of streams and rivers, the gross erosion rates for stream NU3 and river N1 were compared. The average gross erosion for NU3 ($\sim$4,000 tons·km$^{-2}$·yr$^{-1}$) was larger than that of N1 ($\sim$2,400 tons·km$^{-2}$·yr$^{-1}$), and the SD values of N1 and NU3 were 71 and 193 tons·km$^{-2}$·yr$^{-1}$, respectively. To identify their differences, the erosion map, satellite, and aerial imagery were compared. Most wetlands in South Korea are alluvial features located near channels and are frequently inundated during flood events (Fig. 6 (b)). The geospatial analyses also indicated many wetland

and vegetated floodplain regions near the alluvial rivers, with sand as the primary underlying substance (Fig. 6 (c) and (d)). In contrast to the river, the stream watershed did not contain many wetland and flood plains (Fig. 7 (c)). Gravel- and cobble-bed streams were commonly found in moderately steep mountain valleys (Fig. 7 (b)), which were subsequently deposited when water flow rates decreased (Fig. 7 (d)). Thus, the results suggest that stream watersheds carry more sediment, and alluvial rivers provided more opportunities for deposition.

**3.3 Model validation using the sediment delivery ratio**

The regression models developed in this study were validated using the SDR of the gross erosion levels measured at different resolutions, and predicted SD of 14 ungauged watersheds (Fig. 1 (b)). Boyce (1975) suggested that sediment yield varied inversely with the watershed area, whereas other research has suggested a range of SDRs from experimental data (Madiment, 1993; Julien, 2010). The calculated SDRs from gauged rivers, streams, reservoirs, and ungauged watersheds, as well as results

from other studies, are shown in Fig. 8.

The gross erosion results from 3 different resolutions are delineated as bar graphs and indicated that estimated SDR levels were higher in the stream watersheds than rivers. The stream watersheds with values outside the predicted intervals (NU5 and SU3) likely arose from measurement errors associated with the abundance of bushes and trees at gauging points. Most of the

calculated SDRs for ungauged watersheds using multiple regression analyses were within the range of external references; however, some of the calculated SDRs from model-tree data mining were also out of range. This result suggests that the data mining technique could be beneficial for predicting sediment yield under various hydrogeomorphic conditions. Alternatively, since sediment transport is a stepwise process, the proposed multiple regression model based on the USLE structure displayed better validation results. Further, the lowest elevation variable ($LE$) was well correlated with the observed differences between

streams and rivers. However, it did not maintain strong explanatory power for predicting sediment yield compared to the slope of the hypsometric curve.

**4 Discussion**

**4.1 Parameters of erosion and sedimentation**

Relationships between the significant parameters of the proposed model related to land use and estimated SDR were analyzed

(Fig. 9). The negative relationship between the percent of wetland and water cover indicated that these areas provide opportunities for sediment deposition. Additionally, all SDR values (streams, rivers, and reservoirs) increased with increasing urbanized area, albeit to different extents for streams and rivers. Although increase in urbanized areas is not the main cause of soil loss, it can increase opportunities for soil loss. Further, urbanized areas, such as paved roadways and parking lots, can transport more soil particles at increasing flow rates. The hypsometric curve was expressed as the normalized cumulative area,

and height of the watershed outlet, and correlates to flood responses, soil erosion, and sedimentation processes (Langbein, 1947; Strahler, 1952). The hypsometric curves for the 76 watersheds show strong differences in relief aspects between streams, rivers, and reservoir watersheds (Fig. 10 (a)). The results of this study suggest a counter-intuitive relationship between the absolute slope value of the hypsometric curve and the SDR. This may be explained by the topographic irregularity of South





Korea and its effects on erosion and sedimentation processes, where steep mountain watersheds are located far from gauging
stations and floodplains are prone to erosion from the urbanization processes.

**4.2 Effects of spatial resolution**

Geospatial analyses were conducted for upland erosion at three different spatial resolutions, using satellite and aerial imagery
to reveal the mechanisms of the meaningful parameters in the proposed model and other possible factors related to land use.
The 5 x 5 m resolution erosion map showed a significantly improved delineation than that of the lower resolutions (Fig. 11).


As upland areas tend to be more rural croplands with high slope gradients, the corresponding stream watersheds would have a
very high gross erosion rate compared to that of rivers (Fig. 11 (a)). The exposed soils of urbanization are prone to erosion,
potentially releasing abundant quantities of sediment through flooding events. They can also be easily transported to waterways
through surface runoff. Once construction is finished, these locations should change to low erosional risk sites; however, the
urbanization process is nearly continuous. The results also delineated a swale located near the flat regions of the alluvial river
(Fig. 11 (b)). It was also shown that wetlands are not a source of sediment yield; however, agricultural reservoirs are the main
source of sediment deposition, as the annual soil loss for reservoirs was near 0~1 tons·km$^{-2}$·yr$^{-1}$. The erosion map at high
resolutions also accurately captured reservoirs, with increasing variability as resolution decreased (Fig. 11 (c)). Thus, wetlands
and water land cover with low gross erosion rates (0–10 tons·km$^{-2}$·yr$^{-1}$) could be incorrectly distorted as an erosional source
(500–2,500 tons·km$^{-2}$·yr$^{-1}$) at a resolution of 90 m (Fig. 11 (d)).

Additionally, only the high-resolution map (> 30-m resolution) was capable of delineating high-risk erosion areas, such as
meandering channels without bank protection (Fig. 12 (a), white circle). For the development of the USLE, the unit plot had a
length of 22.1 m, width of 1.83 m, and slope declination of 9% (Wischmeier and Smith, 1965). Similar to these reference
conditions, the erosion maps of resolution 5 m and 30 m could be adopted to identify specific sites most in need of erosion and
sediment management. The estimated mean annual soil loss for each land use was calculated for the different resolutions, and
the results are shown in Fig. 13.

The mean annual soil loss of the river watersheds, including many wetlands and water bodies, was more distorted than stream
watersheds; urbanized areas also showed similar results. Over bare land, potentially a main source of sediment release, the
gross erosion displayed the most severe distortion (≤ 100%). In conclusion, the high-resolution erosion maps could be used to
capture areas with high erosion risk (e.g., upstream croplands and construction sites) and detailed features affecting
sedimentation (e.g., wetlands and agricultural reservoirs).

**5 Conclusion**

In this study, the specific degradation (SD) of 62 river gauging stations and 14 reservoirs were investigated across South Korea.
The SD estimated was the highest in reservoirs and calculated SD using a modified Einstein procedure ranged from 1,000
tons·km$^{-2}$·yr$^{-1}$ in upstream mountain watersheds to 100 tons·km$^{-2}$·yr$^{-1}$ in large rivers and as low as 10 tons·km$^{-2}$·yr$^{-1}$ in
downstream reservoirs and alluvial plain rivers. As expected, erosion occurred primarily upstream, and sedimentation took
place in the downstream reservoirs and flood plains. Because the existing regression equation based on the 28 watersheds
located in an alluvial river of South Korea displayed poor accuracy when predicting upstream SD values of the reservoirs (i.e.,
RMSE ~130 tons·km$^{-2}$·yr$^{-1}$), two empirical models using river and stream sediment measurement data were put forth. The
proposed model derived from multiple regression analysis, based on the USLE structure, included the following meaningful
variables: (1) watershed area, (2) mean annual precipitation at gauging stations, (3) percent urbanized area, (4) percent wetland


and water cover, and (5) slope of hypsometric curve parameters. The second data mining model tree contained: (1) mean

annual precipitation at gauging stations, (2) percent of sand at effective soil depth of 0–10, (3) percent of urbanized area, (4) percent water cover, and (5) lowest elevation of the watershed. The two models maintained satisfactory RMSE values of 89 and 59 tons·km$^{-2}$·yr$^{-1}$. Additionally, remote sensing erosion maps (satellite and aerial imagery) employing RUSLE were created at different spatial resolutions. To test model accuracy, the SDR for 14 ungauged watersheds, the sediment yield from the developed models and the gross erosion were also calculated. The erosion-prone areas, such as the cropland and construction

site at the hillslope, could provide abundant sediment and could be deposited on reservoirs and wetlands, which are located near alluvial rivers during flood events. These features were often not visible, and mapping of high-erosion areas was very difficult to delineate on low-resolution maps. When comparing the average annual soil losses on the erosion maps at 5-m and 90-m resolutions, the differences were as large as 100%. The calculated SDRs based on the multiple regression model were within the range suggested by the literature, and supporting the model's use for predicting sediment yield of ungauged

watersheds in South Korea; however, some of the estimated SDRs values from the data mining model were outside the range, indicating that even the data-driven method is limited. Empirical model development using multiple regression analyses and data mining techniques has shown promise for identifying at-risk watersheds. These methods should be applied with caution, because erosion and sedimentation are highly complex physical processes. In conclusion, priority should be given to understanding spatially varied erosion and sedimentation processes under different conditions and focusing in on variable

mechanisms to develop empirical or statistical models. Also, a geospatial analysis with high-resolution data should identify the specific location which require sustainable management. The suggested methodologies here can be utilized for erosion and sediment management and to help understand the mechanisms of these processes in South Korea.





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


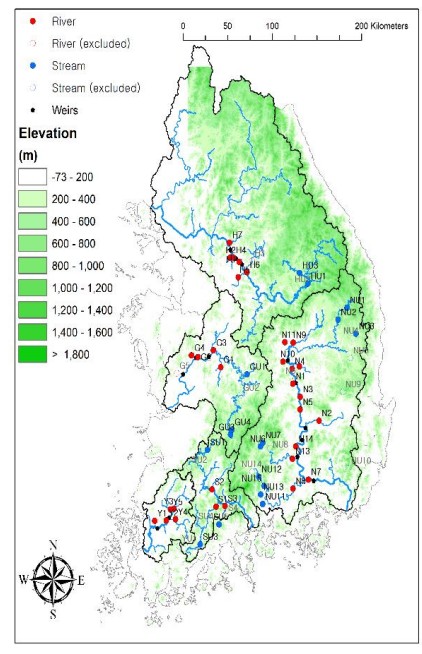

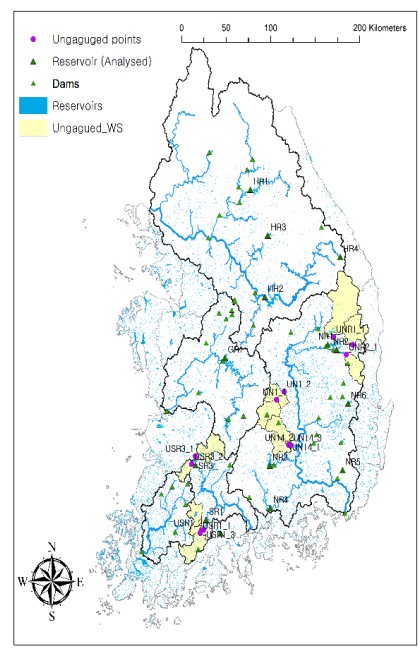

(a)                                        (b)

Figure 1: Study site locations throughout South Korea.



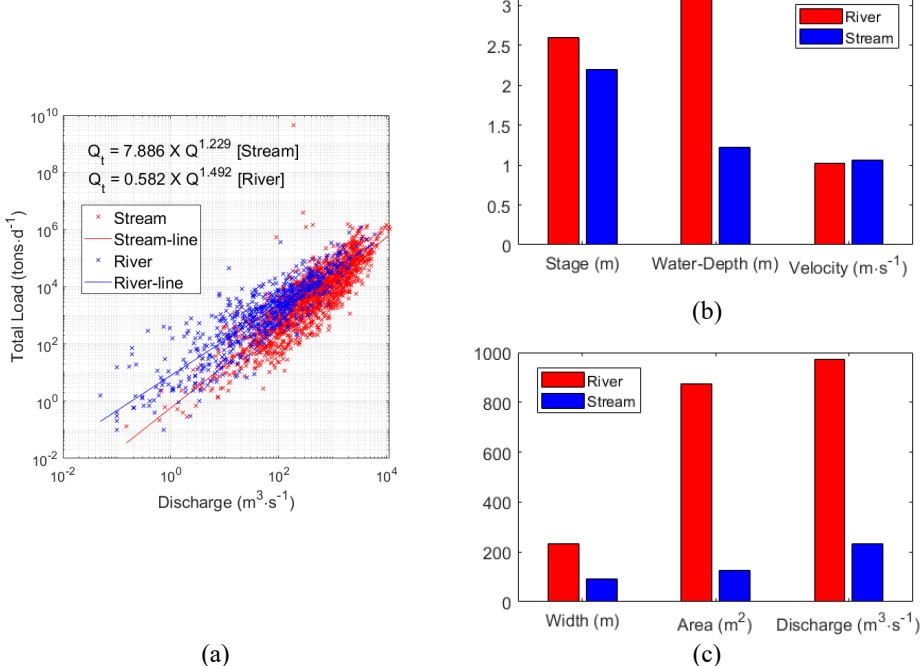

**Figure 2: (a) Sediment rating curve, and (b & c) channel characteristics between streams and rivers of South Korea.**


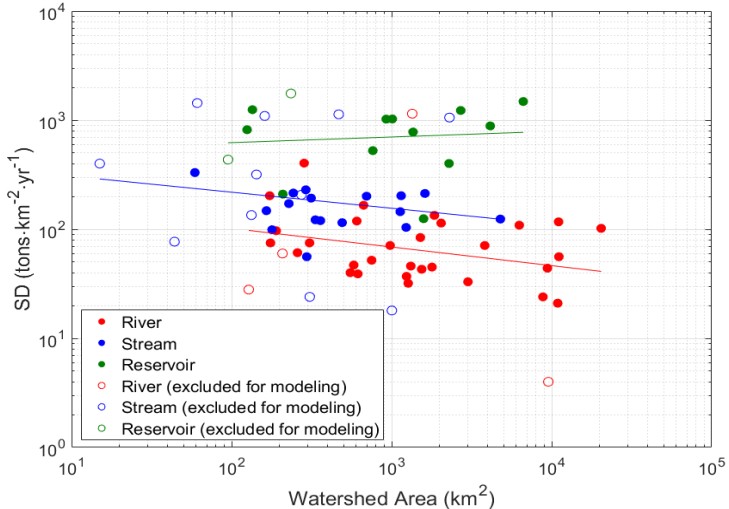

**Figure 3: Specific degradation by watershed area for the analyzed rivers, streams and reservoirs of South Korea**





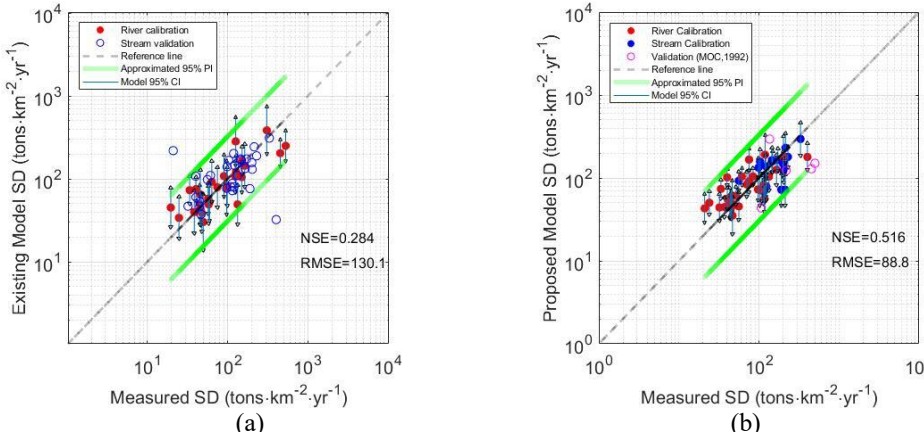

**Figure 4: Prediction of specific degradation from the (a) existing model, and (b) proposed model, using multiple regression analyses.**

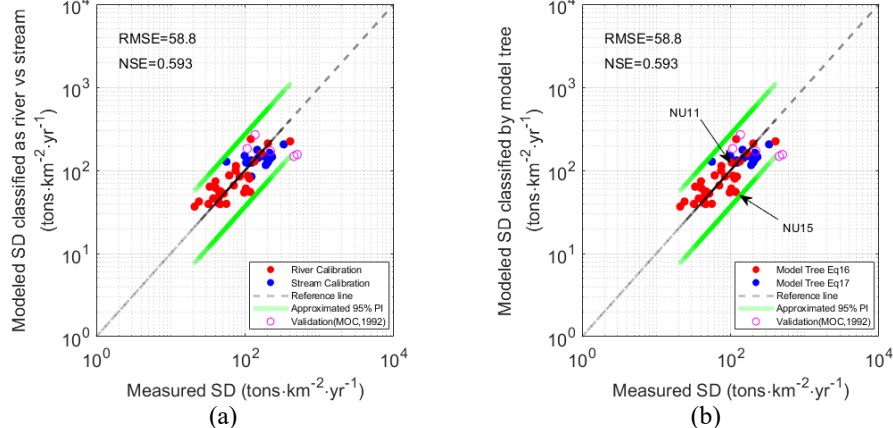

**Figure 5: Prediction of specific degradation from the proposed model using data mining: (a) rivers vs. streams, and (b) Model Tree conditions Eq 16 vs. Eq 17.**





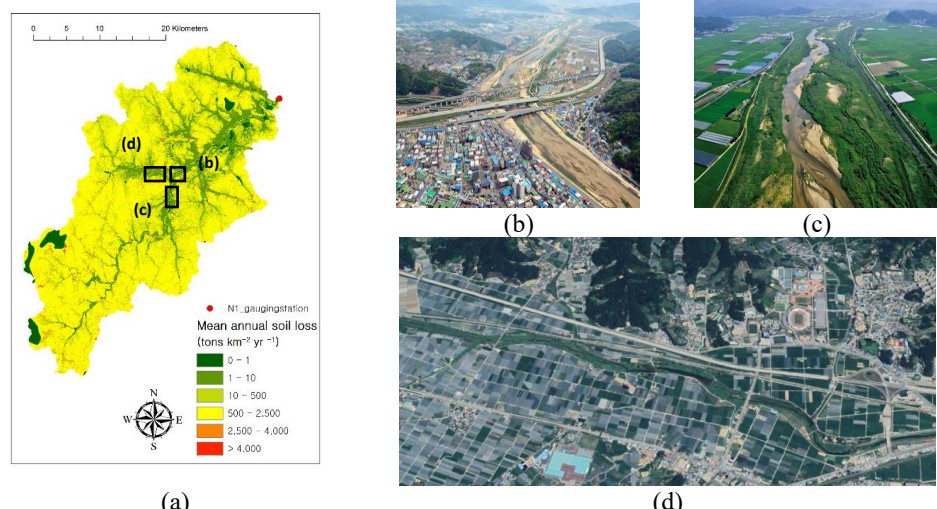

Figure 6: (a) Erosion map, (b–c) aerial photographs, and (d) satellite imagery of river watersheds, ESRI 2020 (study site: N1).


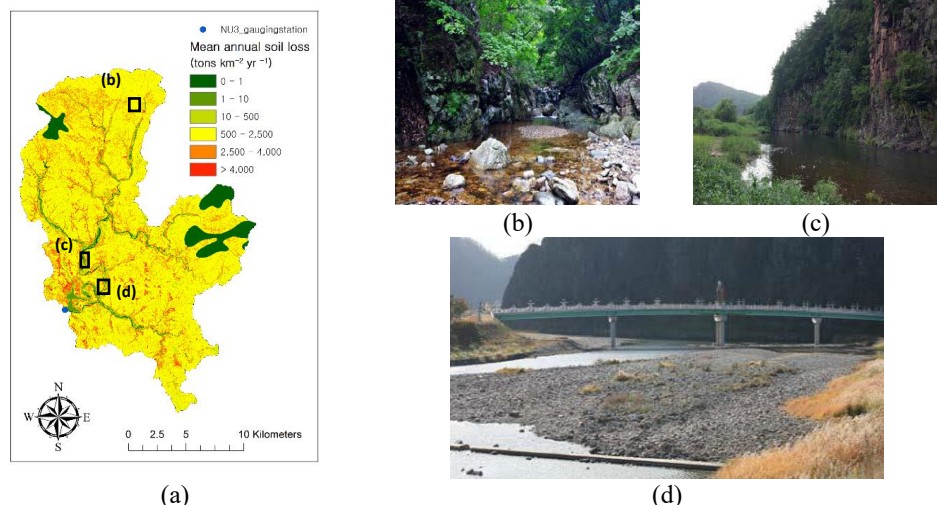

Figure 7: (a) Erosion map, and (b–d) aerial images of stream watershed (study site: NU3).





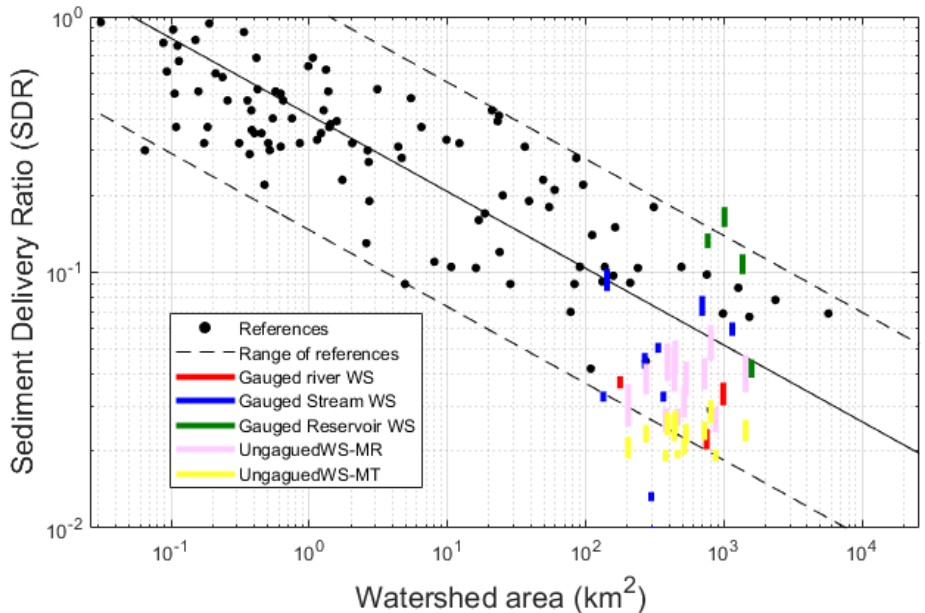

**Figure 8: Sediment delivery ratio for streams, rivers, reservoirs, and ungauged watersheds.**


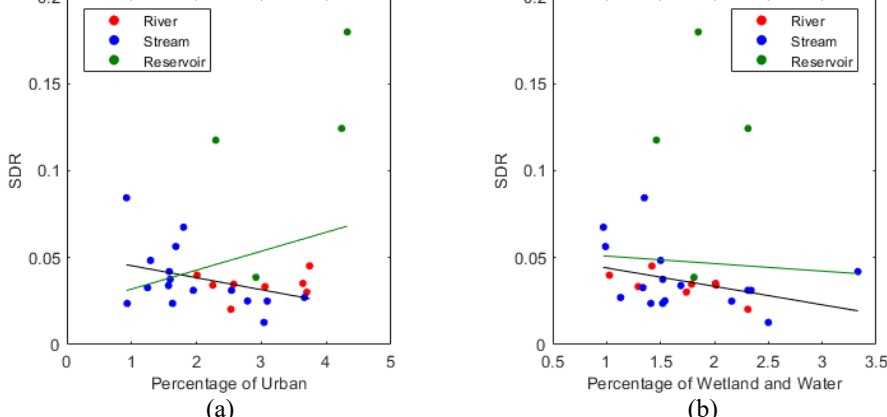

|     |     |
| :-: | :-: |
| (a) | (b) |

**Figure 9: The relationship between the sediment delivery ratio and: (a) the percentage of wetland and water, and (b) the percentage of urbanized area for the rivers, streams, and reservoirs analyzed (black line is the fit of rivers and streams, green line is the fit of rivers, streams and reservoirs).**





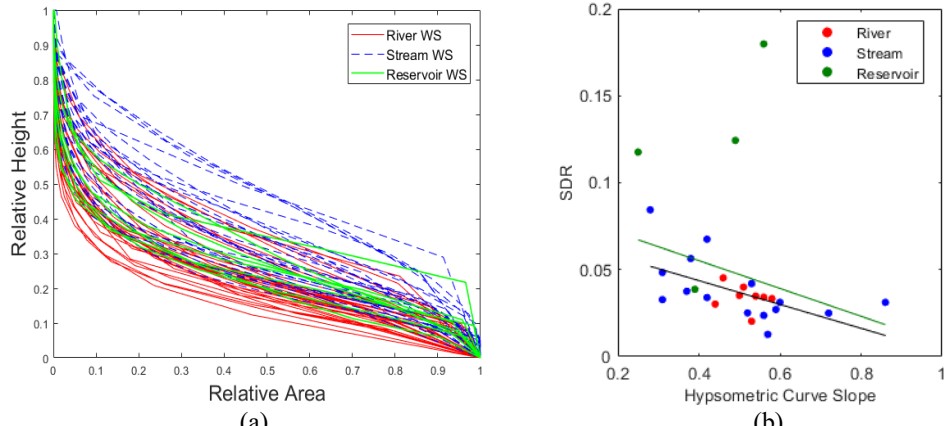

(a)                                                                      (b)

**Figure 10: (a) The hypsometric curve for all river, stream, and reservoir watersheds analyzed, and (b) the corresponding relationship between the sediment delivery ratio and the slope of hypsometric curve (black line is the fit of rivers and streams, green line is the fit of rivers, streams and reservoirs).**






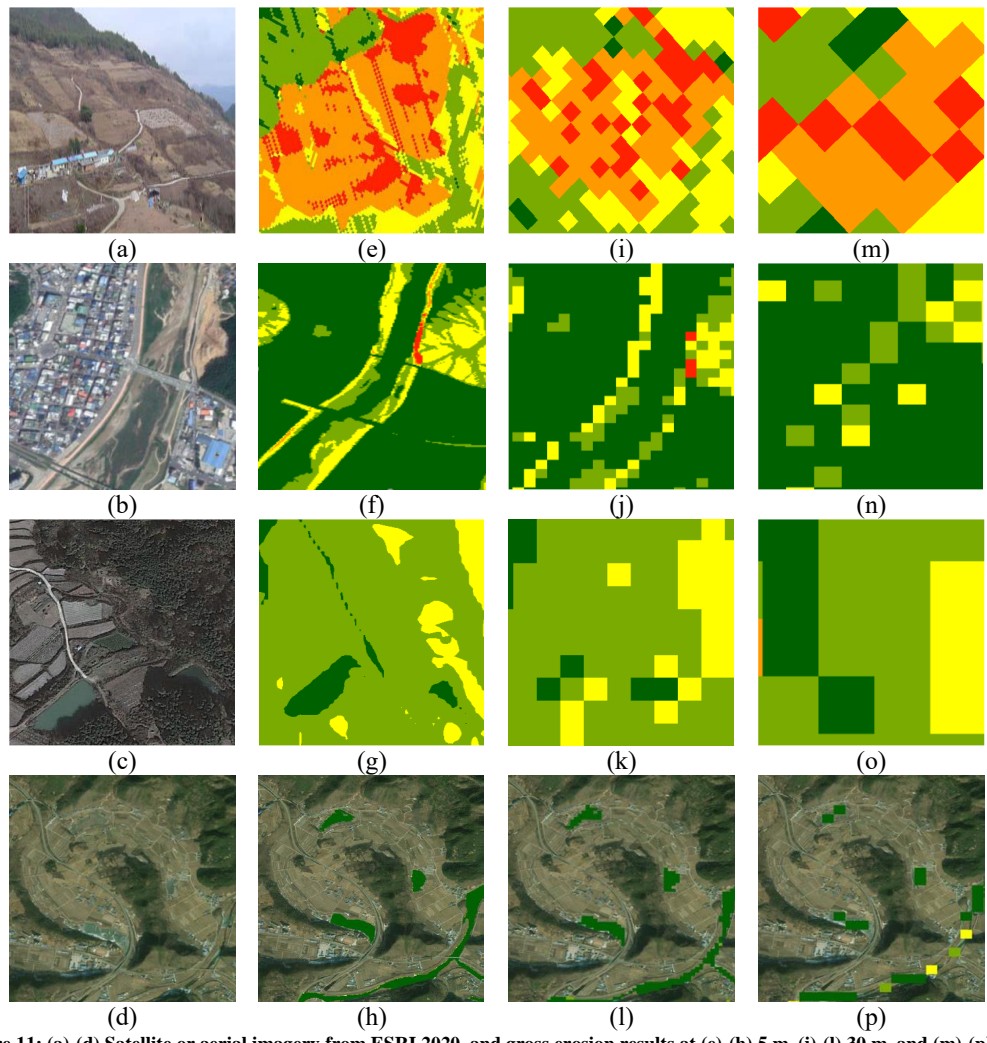

**Figure 11: (a)-(d) Satellite or aerial imagery from ESRI 2020, and gross erosion results at (e)-(h) 5 m, (i)-(l) 30 m, and (m)-(p) 90 m resolution (erosion maps have same scale as Figures 6 and 7).**

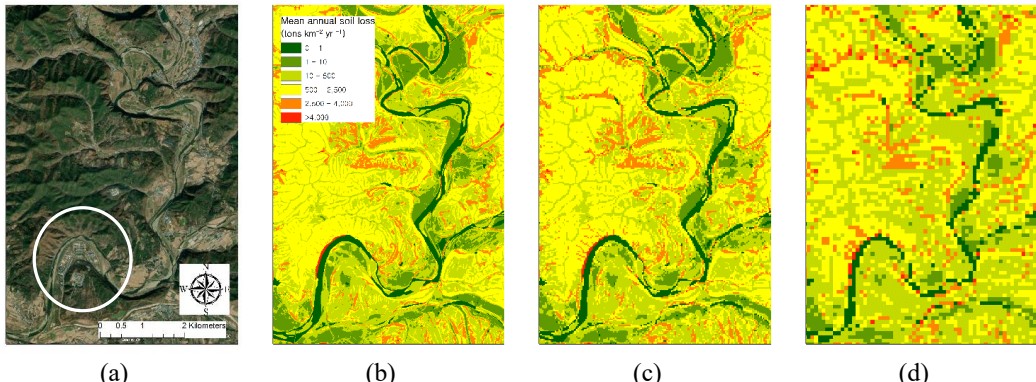

**Figure 12: (a) Satellite image from ESRI 2020, and gross erosion results at (b) 5 m (c) 30 m, and (d) 90 m resolution.**



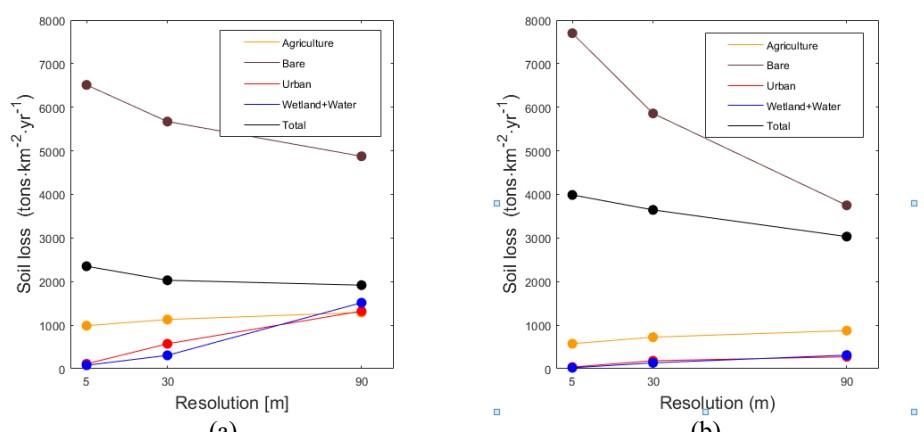

Figure 13: Resolution effects on gross erosion for each land use type for (a) N1, and (b) NU3.


Table 1: The specific degradation of the major reservoirs in South Korea.

| Reservoir (ID) | Year | Area (km²) | Total sediment (10⁶ m³) | Sediment deposition rate (m³·km⁻²·yr⁻¹) | Dry mass density (tons·m⁻³) | Specific Degradation (tons·km⁻²·yr⁻¹) |
|---|---|---|---|---|---|---|
| Soyangriver (HR1) | '06 | 2,703 | 81.5 | 914 | 1.29 | 1,228 |
| Chungju (HR2) | '07 | 6,648 | 130.5 | 853 | 1.67 | 1,484 |
| Hoengseong (HR3) | '13 | 209 | 0.5 | 183 | *1.1 | 210 |
| Gwangdong (HR4) | '12 | 125 | 0.9 | 714 | *1.1 | 818 |
| Andong (NR1) | '08 | 1,584 | 5.5 | 109 | *1.1 | 125 |
| *Imha (NR2) | '07 | 1,361 | 5.6 | 300 | *1.1 | 344 |
| Hapcheon (NR3) | '12 | 925 | 19 | 893 | 1.1 | 1,023 |
| Namriver (NR4) | '04 | 2,285 | 12.5 | 350 | *1.1 | 401 |
| *Miryang (NR5) | '13 | 95 | 3.8 | 380 | *1.1 | 435 |
| *Yeongcheon (NR6) | '05 | 235 | 9.4 | 1,534 | *1.1 | 1,758 |
| Daecheong (GR1) | '06 | 4,134 | 81.4 | 616 | 1.38 | 886 |
| Juam_Main (SR1) | '03 | 1,010 | 5.0 | 469 | 2.1 | 1,026 |
| Juam_reg (SR2) | '03 | 135 | 2.1 | 1,089 | *1.1 | 1,248 |
| Seomjinriver (SR3) | '83 | 763 | 19.0 | 459 | *1.1 | 526 |

* Estimated sediment deposit rate is the same as the design value (i.e., unreliable results).




**Table 2: Estimated specific degradation and data information of all gauging stations.**

| Station | Name | Water shed | Area (km²) | # of Q | # of Sediment | Annual sediment load (ton·yr⁻¹) | Specific degradation (tons·km⁻²·yr⁻¹) |
|---|---|---|---|---|---|---|---|
| H1 | Yeoju | | 11,047 | 3,256 | 97 | 1,295,000 | 117 |
| H2 | Heungcheon | | 284 | 2,832 | 26 | 114,000 | 404 |
| *H3 | Munmak | | 1,346 | 3,213 | 48 | 1,543,000 | 1,147 |
| H4 | Yulgeuk | | 173 | 730 | 29 | 35,000 | 203 |
| H5 | Namhanriver | Han | 8,823 | 1,084 | 30 | 207,000 | 24 |
| H6 | Heukcheon | | 307 | 2,148 | 37 | 23,000 | 75 |
| HU1 | Yeongchun | | 4,782 | 3,633 | 33 | 582,000 | 124 |
| *HU2 | Samok bridge | | 2,298 | 1,437 | 22 | 2,586,000 | 1,057 |
| HU3 | Yeongwol1 | | 1,615 | 2,530 | 27 | 352,000 | 213 |
| N1 | Seonsan | | 979 | 2,878 | 67 | 69,000 | 71 |
| N2 | Dongchon | | 1,541 | 1,430 | 44 | 67,000 | 43 |
| N3 | Gumi | | 10,913 | 1,774 | 33 | 229,000 | 21 |
| N4 | Nakdong | | 9,407 | 2,517 | 53 | 413,000 | 44 |
| N5 | Waegwan | | 11,101 | 2,136 | 147 | 622,000 | 56 |
| *N6 | Ilseon bridge | | 9,533 | 1,826 | 14 | 39,000 | 4 |
| N7 | Jindong | | 20,381 | 3,275 | 84 | 2,087,000 | 102 |
| N8 | Jeongam | | 2,999 | 3,287 | 74 | 100,000 | 33 |
| N9 | Hyangseok | | 1,512 | 1,809 | 63 | 127,000 | 84 |
| N10 | Dongmun | | 175 | 1,826 | 29 | 13,000 | 75 |
| N11 | Jeomchon | Nak dong | 615 | 2,922 | 48 | 24,000 | 39 |
| N12 | Yonggok | | 1,318 | 2,900 | 15 | 61,000 | 46 |
| N13 | Jukgo | | 1,239 | 2,908 | 69 | 46,000 | 37 |
| N14 | Gaejin2 | | 750 | 3,242 | 57 | 39,000 | 52 |
| NU1 | Socheon | | 697 | 2,557 | 15 | 140,000 | 201 |
| NU2 | Yangsam | | 1,147 | 1,789 | 31 | 232,000 | 203 |
| NU3 | Yeongyang | | 314 | 2,191 | 34 | 61,000 | 193 |
| *NU4 | Dongcheon | | 143 | 1,096 | 12 | 45,000 | 318 |
| *NU5 | Cheongsong | | 308 | 3,287 | 12 | 7,000 | 24 |
| NU6 | Geochang1 | | 228 | 2,556 | 31 | 39,000 | 172 |
| NU7 | Geochang2 | | 179 | 2,556 | 19 | 18,000 | 99 |
| *NU8 | Jisan | | 161 | 3,287 | 14 | 176,000 | 1,093 |
| *NU9 | Gohyeon | | 15 | 730 | 26 | 6,000 | 400 |
| *NU10 | Daeri | | 61 | 3,286 | 4 | 87,000 | 1,434 |
| NU11 | Changchon | | 334 | 1,826 | 18 | 41,000 | 122 |
| NU12 | Sancheong | | 1132 | 2,921 | 31 | 164,000 | 145 |
| NU13 | Taesu | | 243 | 1,461 | 16 | 52,000 | 215 |
| *NU14 | Imcheon | | 467 | 3,652 | 14 | 528,000 | 1,132 |
| NU15 | Oesong | | 1232 | 730 | 16 | 128,000 | 104 |
| G1 | Hoedeok | | 606 | 2,902 | 50 | 72,000 | 119 |
| G2 | Gongju | | 6275 | 2,891 | 105 | 682,000 | 109 |
| G3 | Hapgang | | 1850 | 2,192 | 30 | 247,000 | 134 |
| G4 | Useong | | 258 | 730 | 21 | 16,000 | 61 |
| *G5 | Guryong | Geum | 208 | 2,556 | 7 | 12,000 | 60 |
| GU1 | Cheongseong | | 491 | 3,266 | 53 | 56,000 | 115 |
| *GU2 | Hotan | | 1003 | 2,192 | 9 | 18,000 | 18 |
| GU3 | CheonCheon | | 291 | 2,922 | 52 | 67,000 | 230 |
| GU4 | Donghyang | | 165 | 2,556 | 21 | 24,000 | 148 |
| Y1 | Hakgyo | | 190 | 2,921 | 40 | 19,000 | 97 |
| Y2 | Naju | | 2039 | 2,532 | 109 | 233,000 | 114 |
| Y3 | Mareuk | Yeongsan | 668 | 3,269 | 36 | 111,000 | 166 |
| Y4 | Nampyeong | | 580 | 1,422 | 80 | 27,000 | 47 |
| Y5 | Seonam | | 552 | 3,634 | 68 | 22,000 | 40 |





| | | | | | | | |
|---|---|---|---|---|---|---|---|
| *YU1 | Bongdeok | | 44 | 3,648 | 8 | 3,000 | 77 |
| S1 | Jukgok | | 1269 | 3,264 | 15 | 41,000 | 32 |
| S2 | Gokseong | | 1788 | 3,274 | 15 | 80,000 | 45 |
| S3 | Gurye2 | | 3818 | 3,640 | 102 | 172,000 | 71 |
| *S4 | Yongseo | Seom jin | 128 | 1,096 | 14 | 4,000 | 28 |
| SU1 | Gwanchon | | 359 | 3,287 | 44 | 43,000 | 120 |
| *SU2 | Ssangchi2 | | 133 | 1,095 | 14 | 18,000 | 135 |
| SU3 | Gyeombaek | | 295 | 3,277 | 56 | 17,000 | 56 |
| *SU4 | Jangjeon2 | | 273 | 2,191 | 11 | 57,000 | 207 |
| SU5 | Songjeon | | 59 | 2,556 | 33 | 20,000 | 331 |

\* Discarded and/or unreasonable results.

**Table 3: Variables considered in the empirical model.**

| Category | Variables (number of variables) |
|---|---|
| *Geomorphological factors* | Line: total, main, tributary stream length, three stream orders (6)<br><br>Area: watershed area, drainage density, length factor, shape factor (4)<br><br>Relief: average watershed slope, river slope, middle relative height at middle relative area, elevation at middle relative area, lowest elevation, middle elevation, and three hypsometric indices (9) |
| *Climatology factors* | Precipitation at gauging station, averaged value over basin area, rainfall erosivities at gauging station, and averaged value over basin area (4) |
| *Anthropogenic factors (i.e. Land use)* | Percentage of urban, agricultural, forest, pastoral, bare, wetland, water area, and wetland and water area landcover (8) |
| *Pedological Factors* | Percentage of sand, clay, silt, and rock at 0–10, 10–30, 30–50, 0–30, and 0–50 cm effective soil depths (20) |

Table **4**: **Cropping management factor (revised after ME, 2012).**

| Major category | Sub category | C- value | Reference |
|---|---|---|---|
| Urban | Residential<br>Industrial<br>Commercial<br>Recreational<br>Transportation<br>Institutional | 0.01 | (Kim, 2006) |
| Agriculture | Paddy field | 0.1 | (ME, 2012) |
| | Farm | 0.3 | (ME, 2012) |
| | Vinyl Greenhouse | 0.1 | Assumption |
| | Orchard | 0.09 | (ME, 2012) |
| | Generic | 0.2 | Assumption |
| Forest | Deciduous forest<br>Evergreen forest<br>Mixed forest | 0.05 | (ME, 2012) |
| Pasture | Natural pasture<br>Golf<br>Other pasture | 0.15 | (ME, 2012) |
| Wet land | Inland wetland<br>Coast wetland | 0 | (Kim, 2006) |
| Bare land | Mining site<br>Other bare land | 1 | (ME, 2012) |
| Water | Inland water<br>Coast **wetland** | 0 | (ME, 2012) |





Table **5**: **Conservation practice factor (revised after ME, 2012).**

| Land use | Slope (%) | P-value |
|---|---|---|
| Bare land | | 1 |
| Paddy field | Slope < 2% | 0.12 |
| | 2–7% | 0.1 |
| | 7–15% | 0.12 |
| | 15–30% | 0.16 |
| | Slope > 30% | 0.18 |
| Farm | Slope < 2% | 0.6 |
| | 2–7% | 0.5 |
| | 7–15% | 0.6 |
| | 15–30% | 0.9 |
| | Slope > 30% | 1 |
| Pasture | | 1 |
| Forest | | 1 |
| Orchard | | 1 |

Table 6: Calculated VIF values for the developed regression model.

| | Var1 | Var2 | Var3 | Var4 | Var5 |
|---|---|---|---|---|---|
| Model | 1.40 | 1.18 | 2.46 | 1.90 | 1.85 |


Table 7: Averaged gross erosion, using different methods and resolutions.

| Water shed | LS method* | Gross erosion (tons·km⁻²·yr⁻¹) | | | Water-shed | LS method* | Gross erosion (tons·km⁻²·yr⁻¹) | | |
|---|---|---|---|---|---|---|---|---|---|
| | | 5 m | 30 m | 90 m | | | 5 m | 30 m | 90 m |
| N1 | 1 | 2,353 | 2,030 | 1,918 | SU1 | 1 | 3,845 | 3,780 | 3,513 |
| | 2 | 2,478 | | | | 2 | 5,050 | | |
| N10 | 1 | 2,133 | 2,040 | 1,913 | SU2 | 1 | 4,118 | 3,808 | 3,400 |
| | 2 | 2,390 | | | | 2 | 5,470 | | |
| N14 | 1 | 3,398 | 3,088 | 2,678 | SU3 | 1 | 4,393 | 4,123 | 4,053 |
| | 2 | 3,630 | | | | 2 | 4,403 | | |
| NU1 | 1 | 2,978 | 2,533 | 2,213 | SU4 | 1 | 4,928 | 4,718 | 4,283 |
| | 2 | 3,513 | | | | 2 | 5,525 | | |
| NU2 | 1 | 3,398 | 2,968 | 2,648 | NR1 | 1 | 3,230 | 3,043 | 2,740 |
| | 2 | 3,868 | | | | 2 | 3,811 | | |
| NU3 | 1 | 3,988 | 3,645 | 3,033 | NR2 | 1 | 4,429 | 4,017 | 3,648 |
| | 2 | 4,103 | | | | 2 | 4,840 | | |
| NU4 | 1 | 3,768 | 3,538 | 3,088 | SR1 | 1 | 2,925 | 3,109 | 3,496 |
| | 2 | 4,583 | | | | 2 | 3,438 | | |
| NU5 | 1 | 2,833 | 2,593 | 2,380 | SR3 | 1 | 6,267 | 5,800 | 5,486 |
| | 2 | 3,593 | | | | 2 | 6,497 | | |

*1, based on UCA; 2, based on FCL.