# Peer review of "Evaluating and developing a model of specific degradation using geospatial analysis for sediment erosion management in South Korea"

_Hydrology and Earth System Sciences, 2021_

## Author Comment (AC1)

**Responses to the Reviewer 1:**

**► Comment 1**

This study proposed a method to predict sediment yields of soil erosion and sedimentation in South Korea. The authors first developed an empirical prediction model of specific degradation based on multiple regression analysis and tree data mining. In addition the also predicted gross erosion based on RUSLE and derived sediment delivery ratios by dividing gross erosion prediction estimated from RUSLE by sediment yield estimated with the empirical prediction model (Eq. 12) in order to validate the results of the empirical model.

I found the paper very difficult to read. To my opinion, it is mainly due to the fact that the objective of the paper is not clearly stated. As it stands, this paper gives the feeling of a confused compilation of the previous papers (Kang et al., 2019 and 2021) without the added value of this new paper being really defined and discussed. This feeling is reinforced by the fact that more than half of the discussion focuses on the effect of spatial resolution on the RUSLE results, although this is not a central objective of the paper.

Another major concern is the use of SDR to validate the empirical model (cf. chapter 3.3 Model validation using the sediment delivery ratio) when the authors do not have measured SDR data (nor gross erosion references). To my opinion, the use of SDR data from the literature does not allow any conclusion to be drawn on the validation of the sediment yield prediction (or specific degradation) model, especially as these SDR values are known to be strongly linked to hydrosedimentary connectivity of the considered watershed, and therefore very site-dependent (see for instance De Vente et al., 2007, DOI: 10.1177/0309133307076485). As a consequence, there is little justification for the paper's general approach of articulating an empirical catchment model and a RUSLE approach to derive SDR values...

In addition, the discussion has to be completely rewritten in relation to the objective of the paper.

**Response** ◀

We really appreciate the editor's and reviewer's effort in evaluating our manuscript. Your comments were extremely helpful in improving our paper. Following the editor's and reviewer's suggestions, we will conduct thorough revision, and the point-by-point responses to each comment and suggestion are addressed below.

As you suggested, the revised version will provide a clear description about the objective of our study. Indeed, this paper is different from the previous papers (Kang et al., 2019 and 2021). Specifically, the previous study (Kang et al., 2019) discussed the model through a multiple regression analysis with the estimated degradation from the watershed located in alluvial plain river. Moreover, another paper (Kang et al., 2021) focuses on the geospatial analysis of the proposed model using data mining. In this study, additional field measurement data were obtained from mountainous watershed and developed the model using multiple regression analysis and data mining technique. Surprisingly, this study also generated models with similar dependent variables compared to previous models. However, we consider that the present study provided meaningful parameters affecting the erosion and sedimentation processes, and new methodology for evaluating the proposed models. I partly agree with your opinion that this paper is difficult to discern. This is because of the lack of some important information in the manuscript as they were presented in the previous papers. Considering your opinion that the objective of the paper is not clearly stated, the objectives of this study are described as follows:

- 1. Developing models for the estimated 62 specific degradation. (It is one of major differences with the previous studies).
- Evaluating the proposed model using various methods. Specifically, a geospatial analysis considering the resolution effects and various factor calculation methods (for L and S factors) are conducted for meaningful parameters. Additionally, the proposed models are validated with the gross erosion and sediment delivery ratio.

For the other major concerns (using SDR for validation), we completely agree with your opinion that SDR values are extremely site dependent. Therefore, there is little justification to

support our opinion that the model using multiple regression analysis provides better results. However, we strongly consider that Fig. 8 validates the proposed model with the SDR from other studies. For your last suggestion, we will provide additional description in the revised paper.

As the 14 ungauged watersheds were also located in the main five watersheds (Fig. 1 (b)), they do not have significantly different watershed characteristics (i.e., they exhibit similar hydro-sedimentary connectivity of the considered watersheds). However, the suggested SDR with the predicted specific degradation using data mining represents a large difference with the SDR from the gauged river and stream. River managers and geomorphologists should use an efficient and simple method for predicting sediment load at ungauged watershed. This methodology could be considered as an evaluation method of the empirical catchment model for the ungauged watershed.

We have revised the manuscript considering these aspects.

**► Comment 2**

**(Specific Comment-1)**

Line 20-21. You use "percent water" and "percent wetland and water" as distinct parameters of the empirical regression analysis. What about the collinearity of these 2 parameters?

**Response <**

Thank you for your comment. In the abstract, we mentioned two parameters "percent water" and "percent wetland and water". Indeed, they are in collinearity condition. However, each parameter was used for different models. "Percent water" was a significant parameter in the model using multiple regression analysis, and "percent wetland and water" was used for the model using data mining. As they were used independently, we did not consider multi-collinearity between two parameters. We have revised the abstract by separating parameters for each model to avoid the confusion.

► Comment 3

**(Specific Comment-2)**

Line 22. Please clarify the sentence "Additionally, erosion maps from the revised universal soil

loss equation (RUSLE) were generated to validate model variables". Which model variables are supposed to be validated by the RUSLE approach?

Thank you for your comment. In this study, the RUSLE was applied to validate the variables for all suggested models using multiple regression analysis and data mining technique. Specifically, the validation of variables was primarily focused on parameters related to land cover (i.e., anthropogenic factors). "Percent water" for data mining model and "percent wetland and water" for regression model, and "percent of urban" for model were primarily discussed through geospatial analysis.

Additionally, the geospatial analysis supports the proposed result of this study that sediment regimes where erosion occurred were primarily upstream, and sedimentation occurred in the downstream reservoirs and flood plains. To avoid confusion and impart clarity, we have revised it.

**► Comment 4**

**(Specific Comment-3)**

Lines 92-93. Finally which values of the trap efficiencies were used in your studies? Why don't you use Heinemann's formula or a similar formula to estimate them (Heinemann, H. G. 1981. A new sediment trap efficiency curve for small reservoirs. Water Resources Bulletin, 17, 825-830.)

**Response <**

**Response**

Thank you for your comment. As mentioned in line 92, the reservoir trap efficiencies for multipurpose dams in South Korea are typically > 96%. Other studies (MOC, 1992) indicated the trap efficiencies of 0.96, which is generally used for various reservoirs with multi-purpose dams in South Korea. Therefore, we used 0.96 as trap efficiencies based on Brune-curve.

As you suggested, using a Heinemann's formula or other similar formula could be interesting research. However, the formula represents a conceptual design model for predicting sediment

trapping performance of small impoundments. We are unsure about applying this formula for the trap efficiencies of reservoir with multi-purpose dams. However, this interesting approach will be tested in future research with additional agricultural reservoir data. Thank you for your valuable idea.

**► Comment 5**

**(Specific Comment-4)**

Line 99-102. A more detailed presentation of the SD assessment procedure is required in this paper. For future works, I particularly suggest to provide a evaluation of uncertainties in your SD estimations, because this evaluation is necessary if you want to show that your new empirical model is better than the previous ones. In a scarce-data context as yours, it can be assumed that the uncertainties on the SD values will be large (as seen in Fig. 2a for the SY/discharge relationship) and that therefore a variety of models can give similar results when considering the uncertainties in the calibration/validation datasets.

Response

Comment 11

► Comment 12

(Specific Comment-11)

Line 216. What is the signification of SA010 in Eq. 16?

Response

Thank you for your valuable comment. The results that the stream watershed in mountainous region could provide more soil particles and carry more sediment, and the alluvial rivers have more possibilities for deposition (wetland and water) are well supported with field measurement data and geospatial analysis (erosion map using the RUSLE, satellite images, and aerial photos).

Although the proposed results seem general facts, we consider that this is useful information. We believe that these results make a significant contribution to the literature because the general fact was well verified with field measurements and geospatial analysis, which has not been conducted yet.

| Line 254. Madiment, 1993 (or Maidment ?)                                                     | has to be added in the final reference list.                                          |
|----------------------------------------------------------------------------------------------|---------------------------------------------------------------------------------------|
|                                                                                              | Response <                                                                            |
| We will add this reference. Thank you for you                                                | r attentive review.                                                                   |
| ► Comment 15                                                                                 | (Specific Comment-14)                                                                 |
| Line 255. In the sentence "as well as results of these other studies and discussed how their | of other studies", please provide the references context is similar, or not, to yours |
|                                                                                              | Response <                                                                            |
| ► Comment 16                                                                                 | ( Specific Comment-15 )                                                        |
| Page 12. Please consider enlearging Fig. 1 as i                                              | t is difficult to read as it stands .   Response                                      |
| Thank you for your valuable suggestion. We w                                                 | vill enlarge the figure.                                                              |
| ► Comment 17                                                                                 | (Specific Comment-16)                                                                 |
| Page 15. In legend of Fig.7, your aerial imag
images                                      | es (in 7 b-d) looks to be terrestrial (or ground)                                     |
|                                                                                              | Response

**(Specific Comment-21)**

Page 21. Several variables in Table 3 are not defined. What is the significance of "Line:total", "Main" ... What is the difference between "Middle relative height at middle relative area", and "elevation at middle relative area", and "middle elevation" ?

**Response**

Thank you for your detailed comment. As the information of 51 variables represents considerable data, we only defined the variables that are used in the proposed model. If you suggest to add all variables, we will consider adding it.

Linear watershed characteristics include parameters that describe stream network. The stream length was analyzed using the Korea Reach File (KRF) version 3 provided by the Ministry of Environments (ME). It was classified into three parts: (1) mainstream length (Main); (2) tributary length (Tri); and (3) total stream length (Total).

In terms of relief aspects,

"Relative height at middle relative area" vs ""Elevation at middle relative area"

Hypsometric analysis is the distribution of surface area with respect to elevation. It has been typically used for calculating hydrologic information because the basin hypsometry is related to flood response, soil erosion, and sedimentation process. The hypsometric curve can represent the hypsometric analysis, and it typically explains the normalized cumulative area and normalized height from the outlet of the watershed outlet.

As shown in Fig. 10 (a), the "relative height at middle relative area" is the normalized height at middle relative area (0.5). "Elevation at middle relative area" represents elevation (no

---

## Author Comment (AC2)

**Responses to the Reviewer 1:**

| ► **Comment 1** | *(General comment)* |

This study proposed a method to predict sediment yields of soil erosion and sedimentation in South Korea. The authors first developed an empirical prediction model of specific degradation based on multiple regression analysis and tree data mining. In addition the also predicted gross erosion based on RUSLE and derived sediment delivery ratios by dividing gross erosion prediction estimated from RUSLE by sediment yield estimated with the empirical prediction model (Eq. 12) in order to validate the results of the empirical model.

I found the paper very difficult to read. To my opinion, it is mainly due to the fact that the objective of the paper is not clearly stated. As it stands, this paper gives the feeling of a confused compilation of the previous papers (Kang et al., 2019 and 2021) without the added value of this new paper being really defined and discussed. This feeling is reinforced by the fact that more than half of the discussion focuses on the effect of spatial resolution on the RUSLE results, although this is not a central objective of the paper.

Another major concern is the use of SDR to validate the empirical model (cf. chapter 3.3 Model validation using the sediment delivery ratio) when the authors do not have measured SDR data (nor gross erosion references). To my opinion, the use of SDR data from the literature does not allow any conclusion to be drawn on the validation of the sediment yield prediction (or specific degradation) model, especially as these SDR values are known to be strongly linked to hydro-sedimentary connectivity of the considered watershed, and therefore very site-dependent (see for instance De Vente et al., 2007, DOI: 10.1177/0309133307076485). As a consequence, there is little justification for the paper's general approach of articulating an empirical catchment model and a RUSLE approach to derive SDR values...

In addition, the discussion has to be completely rewritten in relation to the objective of the paper.

**Response ◄**

We really appreciate the editor's and reviewer's effort in evaluating our manuscript. Your comments were extremely helpful in improving our paper. Following the editor's and reviewer's suggestions, we will conduct thorough revision, and the point-by-point responses to each comment and suggestion are addressed below.

As you suggested, the revised version will provide a clear description about the objective of our study. Indeed, this paper is different from the previous papers (Kang et al., 2019 and 2021). Specifically, the previous study (Kang et al., 2019) discussed the model through a multiple regression analysis with the estimated degradation from the watershed located in alluvial plain river. Moreover, another paper (Kang et al., 2021) focuses on the geospatial analysis of the proposed model using data mining. In this study, additional field measurement data were obtained from mountainous watershed and developed the model using multiple regression analysis and data mining technique. Surprisingly, this study also generated models with similar dependent variables compared to previous models. However, we consider that the present study provided meaningful parameters affecting the erosion and sedimentation processes, and new methodology for evaluating the proposed models. I partly agree with your opinion that this paper is difficult to discern. This is because of the lack of some important information in the manuscript as they were presented in the previous papers. Considering your opinion that the objective of the paper is not clearly stated, the objectives of this study are described as follows:

1. Developing models for the estimated 62 specific degradation. (It is one of major differences with the previous studies).
2. Evaluating the proposed model using various methods. Specifically, a geospatial analysis considering the resolution effects and various factor calculation methods (for L and S factors) are conducted for meaningful parameters. Additionally, the proposed models are validated with the gross erosion and sediment delivery ratio.

For the other major concerns (using SDR for validation), we completely agree with your opinion that SDR values are extremely site dependent. Therefore, there is little justification to

support our opinion that the model using multiple regression analysis provides better results. However, we strongly consider that Fig. 8 validates the proposed model with the SDR from other studies. For your last suggestion, we will provide additional description in the revised paper.

As the 14 ungauged watersheds were also located in the main five watersheds (Fig. 1 (b)), they do not have significantly different watershed characteristics (i.e., they exhibit similar hydro-sedimentary connectivity of the considered watersheds). However, the suggested SDR with the predicted specific degradation using data mining represents a large difference with the SDR from the gauged river and stream. River managers and geomorphologists should use an efficient and simple method for predicting sediment load at ungauged watershed. This methodology could be considered as an evaluation method of the empirical catchment model for the ungauged watershed.

We have revised the manuscript considering these aspects.

► **Comment 2**                                    *(Specific Comment-1)*

Line 20-21. You use "percent water" and "percent wetland and water" as distinct parameters of the empirical regression analysis. What about the collinearity of these 2 parameters?

**Response ◄**

Thank you for your comment. In the abstract, we mentioned two parameters "percent water" and "percent wetland and water". Indeed, they are in collinearity condition. However, each parameter was used for different models. "Percent water" was a significant parameter in the model using multiple regression analysis, and "percent wetland and water" was used for the model using data mining. As they were used independently, we did not consider multi-collinearity between two parameters. We have revised the abstract by separating parameters for each model to avoid the confusion.

► **Comment 3**                                    *(Specific Comment-2)*

Line 22. Please clarify the sentence "Additionally, erosion maps from the revised universal soil

loss equation (RUSLE) were generated to validate model variables". Which model variables are supposed to be validated by the RUSLE approach?

| Response ◄ |

Thank you for your comment. In this study, the RUSLE was applied to validate the variables for all suggested models using multiple regression analysis and data mining technique. Specifically, the validation of variables was primarily focused on parameters related to land cover (i.e., anthropogenic factors). "Percent water" for data mining model and "percent wetland and water" for regression model, and "percent of urban" for model were primarily discussed through geospatial analysis.

Additionally, the geospatial analysis supports the proposed result of this study that sediment regimes where erosion occurred were primarily upstream, and sedimentation occurred in the downstream reservoirs and flood plains. To avoid confusion and impart clarity, we have revised it.

| ► Comment 4 | *(Specific Comment-3)* |

Lines 92-93. Finally which values of the trap efficiencies were used in your studies? Why don't you use Heinemann's formula or a similar formula to estimate them (Heinemann, H. G. 1981. A new sediment trap efficiency curve for small reservoirs. Water Resources Bulletin, 17, 825-830.)

| Response ◄ |

Thank you for your comment. As mentioned in line 92, the reservoir trap efficiencies for multi-purpose dams in South Korea are typically > 96%. Other studies (MOC, 1992) indicated the trap efficiencies of 0.96, which is generally used for various reservoirs with multi-purpose dams in South Korea. Therefore, we used 0.96 as trap efficiencies based on Brune-curve.

As you suggested, using a Heinemann's formula or other similar formula could be interesting research. However, the formula represents a conceptual design model for predicting sediment

trapping performance of small impoundments. We are unsure about applying this formula for the trap efficiencies of reservoir with multi-purpose dams. However, this interesting approach will be tested in future research with additional agricultural reservoir data. Thank you for your valuable idea.

► **Comment 5**                                                    *(Specific Comment-4)*

Line 99-102. A more detailed presentation of the SD assessment procedure is required in this paper. For future works, I particularly suggest to provide a evaluation of uncertainties in your SD estimations, because this evaluation is necessary if you want to show that your new empirical model is better than the previous ones. In a scarce-data context as yours, it can be assumed that the uncertainties on the SD values will be large (as seen in Fig. 2a for the SY/discharge relationship) and that therefore a variety of models can give similar results when considering the uncertainties in the calibration/validation datasets.

**Response ◄**

Thank you for your comment. In fact, South Korea lacks abundant sediment data. However, this study is a comprehensive study of the sediment yield based on recent and reliable sediment measurement data. We totally agree with you that the uncertainties in estimated SD should be considered for the new empirical model. The following methods were performed to consider uncertainties.

Mentioned in line 108

1) Specific degradation from low sample numbers (sediment measurements $\leq$ 15) are discarded while developing the model.
2) Unreasonable specific degradation compared to nearby watersheds are removed before developing the model

Mentioned in line 95

3) The specific degradation of the reservoir from the unreliable data (providing the result as design sediment deposition data) are discarded

Mentioned in 126

4) The confidence intervals for the developed model are suggested (additional information is available in Kang et al., 2019)

As an objective of this study,

5) the proposed models are evaluated through geospatial analysis and SDR
6) the suggested models are validated to existing model with additional specific degradation from other studies

In terms of methodologies (MEP or sediment data),

an appropriate reference for uncertainties of MEP and data could be available in the following reference.

- Yang, C.Y., Julien, P.Y., 2019. The ratio of measured to total sediment discharge. Int. J.Sedim. Res. 34 (3), 262–269.

The following reference could provide details about errors on total sediment load with same data.

- Kang, W. (2019). Geospatial Analysis of Specific Degradation in SOUTH KOREA. 234 p. Dissertation

The results for extreme specific degradation could be generated from low sediment measurements. However, this problem cannot be solved immediately (all available sediment data is used). Furthermore, we considered the conservative methods for estimating the specific degradation to compare with previous empirical models.

► **Comment 6**                                         *(Specific Comment-5)*

Line 109. I am a bit surprised that some gauging stations were removed from this study whereas many of them were used in the previous ones... To what extent could this partly explain the improved quality of the regression?

**Response ◄**

Thank you for your comment. The proposed model developed with same data. There are some mistakes in Table 2, and we will revise them.

In the previous studies, the data was used as below.

1) Kang et al., 2019 → 28/35 specific degradation in river (H3, N6, N12, G5, S1, S2, and S4 were discarded)
2) Kang et a., 2021 → 34/35 specific degradation in river (H3 are discarded)
3) In this study,
   Same specific degradation in river
   18/28 specific degradation in streams (additional data)

Compared to Kang et al., 2021, in the previous study, the multiple regression analysis was not considered. Additionally, the previous study used the entire data including low sediment data. Hence, we are uncertain whether it could improve the quality of the regression.

► **Comment 7**                        *(Specific Comment-6)*

Line 111 (section 2.2). Please consider dividing this section in several sub-sections to make this section easier to read and understand (At least on sub-section for the empirical approach and another for the RUSLE approach). Please also consider to provide more details on the regression model procedure and the parameters tested both for the regression model and the RUSLE approach.

**Response ◄**

Thank you for your suggestion. Following your suggestion, we have divided the Section 2.2. into several sub-sections to improve the readability of the section

- sub-section for the empirical approach
- sub-section for the RUSLE approach.

Additionally, we will provide more details on the regression model procedure.

► **Comment 8**                        *(Specific Comment-7)*

Line 121. What is the signification of SWAT-K here ?

**Response ◄**

Thank you for your comment. Soil type influences soil erosion and sedimentation process. To estimate the percentage of soil type and K factor, the detailed soil map from National Institute of Agriculture Sciences is used in this study. This detailed soil map contains information about 390 soil series. The specific information about the percentage of soil and rock is extracted from soil database from SWAT-K developed from the Korea Institute of Construction Technology.

Overall, SWAT-K provides comprehensive information about soil properties. We will add this information in the revised version.

► **Comment 9**                        *(Specific Comment-8)*

Line 124. Please clarify what you mean by "based on the RUSLE structure" in a more explicit way. The link is not obvious as the RUSLE structure was developed on a plot based scale and the regression model on a watershed-based scale.

**Response ◄**

Thank you for your valuable comment. As you mentioned, Wischmeier and Smith (1965) used annual data from 10,000 test plots from the agricultural areas in the U.S to develop the Universal Soil Loss Equation (USLE). The Revised Universal Soil Loss Equation (RUSLE) upgraded the USLE by focusing on better parameter estimation (Renard et al., 1997). Additionally, various researchers have attempted obtaining more reasonable watershed-based scale results of gross erosion for sediment yield.

In this study, the terminology "based on the RUSLE structure" implies the result of existing model.

$$SD = 2.45 \times 10^{-7} A^{-0.04} P^{1.94} U^{0.61} W^{-0.64} Sa^{1.51} Hyp^{1.84}$$

We considered that this result has similar structure as the RUSLE structure, implying that the meaningful parameters have a relationship with each factor of the RUSLE

$$Gross\ erosion = RKLSCP$$

- A (watershed area) – average soil loss
- P (mean annual precipitation) –R factor
- Sa (percentage of sand at effective soil depths of 0–10 cm) - K factor
- U and W (related to land use) – C and P
- Hyp (slope of the hypsometric curve) – L and S factor

Some terminologies of "RUSLE structure" are misrepresented in the manuscript, and we will revise it.
* * *
**► Comment 10** *(Specific Comment-9)*

Line 200. What is the signification of "W" in eq. 13 ? idem for "Sa" and "Hyp"...

**Response ◄**

Thank you for your comment.
Existing model explained in line 200 was developed in the previous study.

The parameters can be explained as follows: the percentage of sand in the soil (Sa), percentage area covered by wetlands (W), and slope of the converted hypsometric curve (Hyp).

We will add this information in the revised manuscript. Additional details are available in Kang et al., 2019.

Lines 200-201. As far as I can see, the main difference between the previous model (eq. 13) and the proposed new model (eq. 14) lies in the value of the exponent associated with the Hyps parameter (positive in Eq. 13 and negative in Eq. 14). How do you interpret this difference ?

**Response ◄**

Thank you for your insightful comment. First, we would like to apologize for the lack of explanation. "Hyps (in Eq. 14)" represents the slope of the hypsometric curve between 0.2 and 0.8 of relative area (Below Fig. a ).

However, the "Hyp (in Eq. 13)" represents the slope of the logarithmic hypsometric curve. The equation for hypsometric curve is similar with the equation of relative concentration with reference elevation as derived by Rouse (1937). In the previous study, similar conversion of suspended sediment concentration profile was conducted for hypsometric curve and slope of generated results is exported, and the slope of generated results was exported (Fig. b).

As shown in Fig. (c) and (d), the green line represents a reverse relationship between "Hyp" and "Hyps".

Additional details about this are provided in Kang et al. (2019) and Kang (2019).

We will revise the manuscript to avoid confusion between "Hyp" and "Hyps".

[Figure]

(a)

[Figure]

(b)

[Figure]

[Figure]

|     |     |
| --- | --- |
| (c) | (d) |
* * *
Line 216. What is the signification of SA010 in Eq. 16 ?

**Response ◄**

Thank you for this comment. It represents the percentage of sand at effective soil depths of 0–10 cm. (line 224)
* * *
Lines 248-249. "Thus, the results suggest that stream watersheds carry more sediment, and alluvial rivers provided more opportunities for deposition." Do you think this sentence provides useful new information ?

**Response ◄**

Thank you for your valuable comment. The results that the stream watershed in mountainous region could provide more soil particles and carry more sediment, and the alluvial rivers have more possibilities for deposition (wetland and water) are well supported with field measurement data and geospatial analysis (erosion map using the RUSLE, satellite images, and aerial photos).

Although the proposed results seem general facts, we consider that this is useful information. We believe that these results make a significant contribution to the literature because the general fact was well verified with field measurements and geospatial analysis, which has not been conducted yet.

► **Comment 14**                    *(Specific Comment-13)*

Line 254.   Madiment, 1993 (or Maidment ?) has to be added in the final reference list.

**Response ◄**

We will add this reference. Thank you for your attentive review.

► **Comment 15**                    *(Specific Comment-14)*

Line 255. In the sentence "...as well as results of other studies", please provide the references of these other studies and discussed how their context is similar, or not, to yours...

**Response ◄**

Thank you for your suggestion. We will add the reference for SDR range.

► **Comment 16**                    *(Specific Comment-15)*

Page 12. Please consider enlearging Fig. 1 as it is difficult to read as it stands .

**Response ◄**

Thank you for your valuable suggestion. We will enlarge the figure.

► **Comment 17**                    *(Specific Comment-16)*

Page 15. In legend of Fig.7, your aerial images (in 7 b-d) looks to be terrestrial (or ground) images

**Response ◄**

Thank you for your valuable comment. The legend of Fig. 7 will be revised.

► **Comment 18**                                    *(Specific Comment-17)*

Page 16. Change ungagued into ungauged in Fig. 8 and explain what means WS, MR and MT

**Response** ◄

  Thank you for your attentive review. "WS" represents watershed, "MR" represents multiple regression, and "MT" represents model tree.

We will revise it to avoid confusion. Additionally, we will change "ungagued" to "ungauged" in Fig. 8.

► **Comment 19**                                    *(Specific Comment-18)*

Page 18. Please consider adding a legend to Figure 11, or removing Figure 11.

**Response** ◄

Thank you for your comment. We will add a legend for Fig. 11.

► **Comment 20**                                    *(Specific Comment-19)*

Page 19. In Table 1, Have you an explanation for the very high dry mass density 1 for SR1 (value of 2.1).

**Response** ◄

Thank you for your constructive comment. It is field measurement data, and the value was based on the survey report. We considered the site measurement data, and high dry mass density was obtained from the watershed or reservoir characteristics.

► **Comment 21**                                    *(Specific Comment-20)*

Page 20. In Table 2, please explain why the SD values are not similar from those shown in your previous paper (e.g., Kang et al., 2019). And why some of the stations (e.g., NU4 and NU5) are marked as "discarded and/or unreasonable results) but still considered in Table 7.

**Response** ◄

Thank you for your valuable comment. In this study, we double checked the discharge data (compared with other daily discharge data set and H–Q relationship). When the discharge was not reliable, they were removed.

In other references, every SD values are consistent (dissertation, Kang et al., 2021, and this paper have same value).

Additionally, the SD from low sediment samples or unreasonable SD (compared to nearby watershed) were discarded. As replied to comment 6, there are some mistakes in Table 2. The proposed model was based on same data.

► **Comment 22**                                          *(Specific Comment-21)*

Page 21. Several variables in Table 3 are not defined. What is the significance of "Line:total", "Main" ... What is the difference between "Middle relative height at middle relative area", and "elevation at middle relative area", and "middle elevation" ?

**Response ◄**

Thank you for your detailed comment. As the information of 51 variables represents considerable data, we only defined the variables that are used in the proposed model. If you suggest to add all variables, we will consider adding it.

Linear watershed characteristics include parameters that describe stream network. The stream length was analyzed using the Korea Reach File (KRF) version 3 provided by the Ministry of Environments (ME). It was classified into three parts: (1) mainstream length (Main); (2) tributary length (Tri); and (3) total stream length (Total).

In terms of relief aspects,
  "Relative height at middle relative area" vs ""Elevation at middle relative area"
    -   Hypsometric analysis is the distribution of surface area with respect to elevation. It has been typically used for calculating hydrologic information because the basin hypsometry is related to flood response, soil erosion, and sedimentation process. The hypsometric curve can represent the hypsometric analysis, and it typically explains the normalized cumulative area and normalized height from the outlet of the watershed outlet.
      As shown in Fig. 10 (a), the "relative height at middle relative area" is the normalized height at middle relative area (0.5). "Elevation at middle relative area" represents elevation (not normalized) at middle relative area (0.5)

"Middle elevation" is just median value of watershed elevation.

Additional specific information is available in the following reference:

- Kang, W. (2019). Geospatial Analysis of Specific Degradation in SOUTH KOREA. 234 p. Dissertation

Moreover, "middle relative height at middle relative area" will revised as, "relative height at middle relative area"

► **Comment 23**                                                    *(Specific Comment-22)*

Page 22. Please explain why the conservation practice factors (P-value) for each land use type increase with slopes (Table 5). In general, the conservation practice factors do not have a direct systematic link with slopes. Morover, the influence of slopes in RUSLE is already taken into account through the LS factor.

**Response ◄**

Thank you for your comment. There are several methods for estimating practice factors.

In this study, we attempted using official notification "Notification on investigation of soil erosion status" from the Ministry of Environment for using "RUSLE".

The notification is proposed considering South Korea conditions and field survey.

---

## Author Comment (AC3)

**Responses to the Reviewer 2:**

| ► **Comment 1** | *(General comment)* |

The research deals about evaluating specific degradation and sediment yield in South Korea. The procedure is a composed by different stages that considers: 1) analyzing measurements of sediment yield from 62 streams/rivers and 14 reservoirs, 2) developing a regression / tree mining model for sediment yield in 47 up-streams catchments, 3) using RUSLE with mapping variables to validate the model, 4) using 16 ungauged watersheds to validate empirical data, and, 5) remote sensing is used for spatial variables.

We founded the subject of the article interesting, but in general the manuscript is a little confusing on how the procedures are adopted. We had a more clear understanding looking back to the article Kang et al., 2019 where a similar research is conducted and explained in a more fluid behave. We think that the use of methodologies adopted should be follow a more linear explication of the procedures, we are a bit perplexed about the question of spatial resolution that can be interesting but that is to much in evidence in discussions. Some concerns are on the material and methods not really clear or reported from previous article where better explained (e.g. the use of TE, trap efficiency in in defining SD, specific degradation, questionable dealing on both the catchment and the reservoir; use and description of the Modified Einstein Procedure - MEP).

In general, our opinion is that the manuscript could be reconsidered for the publication in HEES Journal after a new submission.

| **Response ◄** |

We really appreciate the editor's and reviewer's effort in evaluating our manuscript. Your comments were extremely helpful in improving our paper. Following the editor's and reviewer's suggestions, we conducted thorough revision, and the point-by-point responses to each comment and suggestion are addressed below.

As you mentioned, this study is a comprehensive study involving previous study results, additional data, and new methodology. Moreover, we agree with your opinion that some important information reported in previous article has not been included. If you require us to add the information, we will briefly introduce them and cite those studies for potential readers.

Additionally, following your suggestion, the manuscript have been rewritten for better understanding.

**► Comment 2 *(Specific Comment-1)**

L19. "significant parameters:": the term significant, should have a statistical meaning.

We think the abstract it is a bit confusing when showing the procedures and the results should be more concise.

**Response ◄**

Thank you for your comment. Both the methods used for developing the model (i.e., multiple regression analysis and data mining technique) are based on statistical theory. As the "significant parameters" were decided based on statistical approach, We consider that the terminology would be valid to use.

Following your suggestion, we will revise the abstract to make it more concise.

**► Comment 3 *(Specific Comment-2)**

L32-41 We are not sure that a pedagogic description of the erosion terms is necessary. In general, the introduction must give a larger spectrum of the state of the art that is much wide than that here showed.

**Response ◄**

Thank you for your constructive comment. We agree with your opinion that "a pedagogic description would be not necessary in the article". However, considering the confusion in the terminology of erosion with other references, we have provided a brief explanation about the erosion term which is used in this article. If it is not necessary, we will consider removing it. Additionally, we will provide more recent references and wide spectrum of the state of the art in the revised manuscript.

**► Comment 4 *(Specific Comment-3)**

L79-81."When water enters a reservoir, the flow velocity decreases, flow depth increases, and sedimentation occurs as a result of the overall decreased transport capacity of the stream.": it is true but, over a pedagogic approach, here not demanded, scientifically thinking, it is a bit more complex.

[Figure]

**Response ◄**

Thank you for your valuable comment. Following your suggestion, the sentence of pedagogic approach will be removed.

**► Comment 5**                                         *(Specific Comment-4)*

L84 sediment deposition rate (ð•‘%ð•‘Ÿ, m3·km-2·yr-1): in this form is not appropriate to call it like this, it is more an erosion/denudation rate in the end (m-1y-1) (i.e., metres over the catchment surface in a period)

**Response ◄**

Thank you for your comment. We agree with your opinion that an erosion/denudation rate is more appropriate. As "erosion/denudation rate" and "sediment deposition rate" have simple relationship, we just provided "sediment deposition rate," which is used in the original form as in survey report.
We will revise it, and sediment deposition rate will be changing as erosion/denudation rate.

**► Comment 6**                                         *(Specific Comment-5)*

L85: field measurements: this is maybe more a continuous monitoring of the dams

[Figure]

**Response ◄**

Thank you for your comment. K-water has conducted a sediment survey every 10 years for multi-purpose and storage dams from impounded water. In the sediment survey, the water elevation and ground level measurements were obtained to estimate the change of reservoir capacity. The reservoir capacity results were from field measurement data. If it persuades confusion, "field measurement" can be changed as "field measurement for continuous monitoring"

**► Comment 7**                                         *(Specific Comment-5)*

L89 impoundment: not sure it is the right term for that

Thank you for your comment. We have used "the measured reservoir capacity from the impoundment of water". It will be changed as "the measured reservoir capacity at first impoundment"

**► Comment 8**                                    *(Specific Comment-7)*

Equations 1 and 2: x: not adapted mathematical notation

**Response ◄**

Thank you for your comment. We will revise it.

**► Comment 9**                                    *(Specific Comment-8)*

Equation 2 (Specific Degradation): $\delta$'$\ddagger\delta$, is the trap efficiency (%): we suppose this is more a fraction than a percentage. If using the term TE, the term Specific Degradation is questionable because is the part of sediment captured by the dam while the degradation of the surface catchment refers to the whole sediment eroded.

**Response ◄**

Thank you for your comment. We will express efficiency as percentage.

As mentioned, the trap efficiency of reservoir is approximately 96%, implying that most of the sediment is captured by the dam. Additionally, the specific degradation of the reservoir provides higher value than stream located upstream of reservoir. Based on this result, we consider that the term "specific degradation" can be used.

**► Comment 10**                                   *(Specific Comment-9)*

Caption Figure2: maybe add that are average values

**Response ◄**

Thank you for your suggestion. We will revise it.

► **Comment 11**                                                     *(Specific Comment-10)*

L100. The Modified Einstein Procedure (MEP) should be at least briefly described and why the author choose this procedure (maybe suitable to this kind of data or study site etc.)

**Response ◄**

Thank you for your comment. We considered that "MEP" is a common method for estimating total sediment load, and the conservative methods for estimating the specific degradation would be required to compare previous empirical models.

We will provide a brief description and reason for using this method.

► **Comment 11**                                                     *(Specific Comment-10)*

L111-190. We suggest to better structure this part to give a more clear presentation of the models, it is a little bit confusing and sometimes not enough well in details for equations. Additionally, some terms are not clear, W versus WW for instance (Eq. 13, 14, 16, 17) or the eq. 15 itself.

**Response ◄**

Thank you for your suggestion. We will divide the Section 2.2. into several sub-sections and add specific information about model development process.

► **Comment 12**                                                     *(Specific Comment-10)*

Is a bit confusing the part of models, showing a sort of evolution of previous models; we think a better structured explication should help.
L250-255. The use of SDR from the literature as in figure 8 is in general allowed, but should pay attention to the specific condition of the reservoirs, being the SDR very dependent on the specific vegetation of the soil (as also observed in this research) and connectivity condition of the reservoirs.

**Response ◄**

Thank you for your suggestion. The proposed models were developed using additional data, and they provide a better predictability. We will add additional structured description about this.

We completely agree with your opinion that SDR values are extremely site dependent. The response for General Comment from Reviewer 1 could sufficiently answer this question. Additionally, the reservoir data were not used for model development. They are provided in Fig. 8 to support the reliability of using SDR data from the literature.

[Figure]

► **Comment 13**                                        *(Specific Comment-12)*

Discussion

Here, we talk about processes (eroding land surface) and a methodological issue, spatial resolution. We are not sure a methodological issue is important, as it is not the core of this paper, while the process has a limited discussion.

[Figure]

**Response ◄**

Thank you for your comment. A methodological issue is not a core of this paper; however, we consider that this section efficiently supports the result of the suggested model and geospatial analysis. Moreover, it provides important results regarding a counter-intuitive relationship (line 277) and sediment characteristics in South Korea (line 279). We will add additional details discussing this process (i.e., limitation of our study, and future work).

[Figure]

► **Comment 14**                                        *(Specific Comment-13)*

Conclusion

All the main finding are evoked, maybe a more synthetic or better structured presentation should help.

[Figure]

**Response ◄**

Thank you for your comment. We will provide a better presentation of our study.

---

## Author Comment (AC4)

**Responses to the Reviewer 3:**

| ► **Comment 1** | *(General comment)* |

There are a number of significant problems with this paper that would need to be addressed before it could be considered for publication, so I think it should be rejected in its current form.

a) Even if one believes that the sediment-delivery ratio is physically meaningful, rather than an artefact of the ways erosion rates have been analyzed historically (Parsons et al. doi: 10.1002/esp.1395), the fact that the paper considers them to be a good test of the modelled rates of erosion is highly problematic. There are common factors in the numerator and denominator of equation (12) that will lead to issues of spurious correlation. The "test" seems to be a comparison of whether the new model can fall somewhere within the bounds of the SDR estimated elsewhere, which is a target of over an order of magnitude. This target is missed in a non-negligible number of cases, and the text then turns to special pleading of why specific datasets are problematic. Either you believe your data or you don't!

b) I do not see the rationale for the structure of the regression model in equation (3). There are many critiques in the literature of the structure of (R)USLE. Furthermore, this is not the same structure, as it is the product of powers of the original variables.

c) Although the "proposed model" has a better RMSE, it seems to have more bias than the other models, overpredicting lower values, and underpredicting higher ones.

d) The description of "data mining" to produce alternative model structures is minimal and wouldn't allow the approach to be replicated. In the results, "meaningful parameters" are mentioned, but it is not clear what meaning they have. In particular, what is the physical meaning of "lowest elevation"?

The overall aim and rationale of the paper is vague. There seems to be an invaluable dataset underlying the paper that could be much better employed in estimated sediment fluxes in different locations.

We really appreciate the editor's and reviewer's effort in evaluating our manuscript. Your comments were extremely helpful in improving our paper. Following the editor's and reviewer's suggestions, we conducted thorough revision, and the point-by-point responses to each comment and suggestion are addressed below.

The major objectives of this study are

- Estimate the specific degradation of South Korean rivers and reservoirs
- Developing an empirical model for specific degradation using recent and large amount of data (entire sediment data in South Korea)
- Evaluating the proposed model through geospatial analysis considering resolution effects; SDR is additionally used for evaluating the suggested model

**(Conclusion)** The suggested methodologies could be utilized for erosion and sediment management to understand the mechanisms of these processes in South Korea.

River managers and geomorphologists should use an efficient and simple method for predicting sediment load. We consider that the suggested models could be useful for South Korean rivers. It could be used for sustainable erosion and sediment management (for gauged watershed). The proposed model could also be used to predict a specific degradation in ungauged watershed in South Korea.

To predict the sediment discharge for ungauged watershed, the method applied in South Korea can be classified into four methods: 1) River sediment data (FD-SRC), 2) Reservoir surveys (sediment deposition), 3) Empirical methods, and 4) sediment discharge at nearby watershed (similar watershed characteristics). Additionally, RUSLE results are occasionally used, when the data for above methods are not available. This study is a comprehensive study involving five methods.

We partly agree with your opinion that considering SDR as a good test could be problematic. However, it is difficult to get data of sediment discharge at the ungauged watershed. However, data of watershed characteristics (could also use for RUSLE) are available.

Furthermore, as the watershed area affects the shapes of the curves of flow duration, water and sediment discharge, it can provide a first approximation for erosion and

sedimentation processes.

These methodologies could be considered an evaluation method of the empirical catchment model for the ungauged watershed.

b) The structure of equation (3) is the most common form of empirical regression model for sediment discharge. (Kang et al., 2019 and 2021 represented many existing regression models, most of them were developed using this form).

Although there are many critiques in the literature regarding the structure of (R)USLE, it has been widely used worldwide to estimate annual soil erosion from hill slope and gross erosion for sediment yield. In terms of terminology of "RUSLE structure," the response for the Comment 9 from Reviewer 1 would be sufficient.

c) We have a different opinion regarding your review that the proposed model have more bias than other models. The existing model which was developed in the previous study (Kang et al., 2019) provided good predictability. It was developed with only 28 SD values for rivers. The existing model exhibited optimal performance for the SD of streams (in mountainous watershed). The proposed model could also provide better accuracy for predicting upstream SD values of the reservoir. In figure 4 (a) and (b), the predicted SD for stream (blue) from the proposed model is higher than the predicted SD for stream from the existing model.

d) We will provide additional details about "data mining" process.

In terms of meaningful parameters, we have provided the reason why they are deemed "meaningful".

(1) drainage area, (2) mean annual precipitation

: They are the common parameters which are used in the empirical model of specific degradation.

(3) percent urbanized area, (4) percent water, (5) percent wetland and water

: We provided the details through the geospatial analysis.

(6) percent sand at effective soil depths of 0–10 cm, (7) slope of the hypsometric curve, and

: In the previous study, we explained the reason why they are considered meaningful parameters. We will briefly introduce them in the revised manuscript.

(8) watershed minimum elevation ("Low elevation")

: The fluvial system can be conceptually classified into three zones (Fig. (a)). Zone 1 represents the erosional zone in upland areas with sediment production into steep bed streams and rivers. Mountain streams flow rapidly through steep slopes in a V-shape valley. In the case of South Korean rivers, numerous upstream mountain headwaters flow directly into bedrock streams. Zone 2 represents a transport zone of water and sediment with long sand-bed river. This study primarily focuses on Zones 1 and 2, and most channels in Zone 1 can be considered as streams and those in Zone 2 can be considered as rivers. The geospatial analysis all delineates it well.

In every watershed, the gauging station, which is the outlet of the watershed, is located at the watershed minimum elevation. Although some gauging stations are in transfer zone between Zones 1 and 2, the low elevations efficiently classified the rivers and streams (Fig. (b)).

We will add this description in the revised manuscript.

[Figure]

(a)                                         (b)

Overall, the revised paper will clearly provide the objective of this paper.